# Adaptive Selective Sampling for Online Prediction with Experts

**Rui M. Castro**
Eindhoven University of Technology,
Eindhoven Artificial Intelligence Systems Institute (EAISI)
rmcastro@tue.nl

**Fredrik Hellström**
University College London
f.hellstrom@ucl.ac.uk

**Tim van Erven**
University of Amsterdam
tim@timvanerven.nl

## Abstract

We consider online prediction of a binary sequence with expert advice. For this setting, we devise label-efficient forecasting algorithms, which use a selective sampling scheme that enables collecting much fewer labels than standard procedures. For the general case without a perfect expert, we prove best-of-both-worlds guarantees, demonstrating that the proposed forecasting algorithm always queries sufficiently many labels in the worst case to obtain optimal regret guarantees, while simultaneously querying much fewer labels in more benign settings. Specifically, for a scenario where one expert is strictly better than the others in expectation, we show that the label complexity of the label-efficient forecaster is roughly upper-bounded by the square root of the number of rounds. Finally, we present numerical experiments empirically showing that the normalized regret of the label-efficient forecaster can asymptotically match known minimax rates for pool-based active learning, suggesting it can optimally adapt to benign settings.

## 1 Introduction

This paper considers online prediction with expert advice in settings where collecting feedback might be costly or undesirable. In the classical framework of sequence prediction with expert advice, a forecasting algorithm aims to sequentially predict a stream of labels on the basis of predictions issued by a number of experts (see, for instance, [1, 2, 3] and references therein). Typically, the forecaster receives the correct label after making a prediction, and uses that feedback to update its prediction strategy. There are, however, situations where collecting labels is costly and potentially unnecessary. In the context of online prediction, this naturally leads to the notion of *selective sampling* strategies, also called *label-efficient prediction* [4, 5, 6, 7, 8, 9]. In this line of work, there is a natural tension between performance (in terms of regret bounds) and label complexity, i.e., the number of labels collected. For a worst-case scenario, the optimal label-efficient strategy amounts to "flipping a coin" to decide whether or not to collect feedback, irrespective of past actions and performance [5]. Indeed, in the worst case, the number of labels that one has to collect is linear in the number of rounds for any algorithm [10]. This is a rather pessimistic perspective, and can miss the opportunity to reduce label complexity when prediction is easy. With this in mind, the adaptive selective sampling algorithms we develop follow naturally from a simple design principle: optimize the label collection probability at any time while preserving worst-case regret guarantees. This principled perspective leads to a general way to devise simple but rather powerful algorithms. These are endowed with optimal worst-case

37th Conference on Neural Information Processing Systems (NeurIPS 2023).

performance guarantees, while allowing the forecaster to naturally adapt to benign scenarios and collect much fewer labels than standard (non-selective sampling) algorithms.

From a statistical perspective, the scenario above is closely related to the paradigm of *active learning* [11, 12, 13, 14, 15, 16]. For instance, in pool-based active learning, the learner has access to a large pool of unlabeled examples, and can sequentially request labels from selected examples. This extra flexibility, when used wisely, can enable learning a good prediction rule with much fewer labeled examples than what is needed in a *passive learning* setting, where labeled examples are uniformly sampled from the pool in an unguided way [17, 14, 15, 18, 19, 20, 21, 22, 23, 24, 25, 26, 27]. Our work is partly motivated by such active learning frameworks, with the aim of devising a simple and adaptive methodology that does not rely on intricate modeling assumptions.

The main contributions of this paper are novel label-efficient exponentially weighted forecasting algorithms, which optimally decide whether or not to collect feedback. The proposed approach confirms, in a sound way, the intuition that collecting labels is more beneficial whenever there is a lack of consensus among the (weighted) experts. The proposed algorithms are designed to ensure that, in adversarial settings, they retain the known worst-case regret guarantees for full-information forecasters (i.e., forecasters that collect all labels) while providing enough flexibility to attain low label complexity in benign scenarios. To characterize the label complexity of the label-efficient forecaster, we focus on a scenario where the expected loss difference between the best expert and all other experts for all $n$ rounds is lower-bounded by $\Delta$, and show that the label complexity is roughly $\sqrt{n}/\Delta^2$, ignoring logarithmic factors. This shows that the label-efficient forecaster achieves the "best of both worlds": it smoothly interpolates between the worst case, where no method can have optimal regret with less than $O(n)$ queries, and the benign, stochastic case, where it is sufficient to make $O(\sqrt{n})$ queries. Finally, to further examine the performance of the label-efficient forecaster, we conduct a simulation study. We find that the performance of the label-efficient forecaster is comparable to its full-information counterpart, while collecting significantly fewer labels. Intriguingly, for a threshold prediction setting studied in [14], the numerical results indicate that the label-efficient forecaster optimally adapts to the underlying prediction problem, so that its normalized regret displays the same asymptotic behavior as known minimax rates for active learning.

Before formally introducing our setting, we discuss additional related work. Selective sampling for online learning was studied by [4, 5, 28], with a focus on probabilistic threshold functions and margin-based sampling strategies. Similarly, [8] consider kernel-based linear classifiers, and base their sampling procedure on the estimated margin of the classifier. For the same setting as we consider, [29, 30] propose a selective sampling approach based on the maximum (unweighted) prediction disagreement among the experts, and numerically demonstrate its merits. Finally, results in a similar spirit to ours have recently been established in different settings. Namely, for a strongly convex loss, [9] devised an algorithm for selective sampling with expert advice, which provably retains worst-case regret guarantees, where the sampling strategy is based on the variance of the forecaster's prediction. [31] study a setting with shifting hidden domains, and establish a tradeoff between regret and label complexity in terms of properties of these domains. For a setting where the hypothesis class has bounded VC dimension and the data satisfies a Tsybakov noise condition, [32] devise a sampling strategy, with bounds on the regret and label complexity, based on a notion of disagreement where hypotheses are discarded based on their discrepancy relative to the empirical risk minimizer.

## 2  Setting

Throughout, we focus on a binary prediction task with the zero-one loss as a performance metric. We refer to $y_t$ as the outcome at time $t \in [n] := \{1, \ldots, n\}$. No assumptions are made on this sequence, which can potentially be created in an adversarial way. To aid in the prediction task, the forecaster has access to the predictions of $N$ experts. The prediction of the forecaster at time $t$ can only be a function of the expert predictions (up to time $t$) and the observed outcomes up to time $t - 1$. Furthermore, the algorithm can make use of internal randomization.

Formally, let $f_{i,t} \in \{0, 1\}$, with $i \in [N] := \{1, \ldots, N\}$ and $t \in [n]$, denote the advice of the experts. At every time $t \in [n]$, the forecasting algorithm must: (i) output a prediction $\hat{y}_t$ of $y_t$; (ii) decide whether or not to observe $y_t$. Specifically, for each round $t \in [n]$:

- The environment chooses the outcome $y_t$ and the expert advice $\{f_{i,t}\}_{i=1}^{N}$. Only the expert advice is revealed to the forecaster.

- The forecaster outputs a (possibly randomized) prediction $\hat{y}_t$, based on all of the information that it has observed so far.

- The forecaster decides whether or not to have $y_t$ revealed. We let $Z_t$ be the indicator of that decision, where $Z_t = 1$ if $y_t$ is revealed and $Z_t = 0$ otherwise.

- A loss $\ell(\hat{y}_t, y_t) := \mathbb{1}\{\hat{y}_t \neq y_t\}$ is incurred by the forecaster and a loss $\ell_{i,t} := \ell(f_{i,t}, y_t)$ is incurred by expert $i$, regardless of the value of $Z_t$.

Our goal is to devise a forecaster that observes as few labels as possible, while achieving low regret with respect to any specific expert. Regret with respect to the best expert at time $n$ is defined as

$$R_n := L_n - \min_{i \in [N]} L_{i,n} \,,$$

where $L_n := \sum_{t=1}^n \ell(\hat{y}_t, y_t)$ and $L_{i,n} := \sum_{t=1}^n \ell(f_{i,t}, y_t)$. Note that the regret $R_n$ is, in general, a random quantity. In this work, we focus mainly on the expected regret $\mathbb{E}[R_n]$.

Clearly, when no restrictions are imposed on the number of labels collected, the optimal approach would be to always observe the outcomes (i.e., take $Z_t = 1$ for all $t \in [n]$). This is optimal in a worst-case sense, but there are situations where one can predict as efficiently while collecting much fewer labels. The main goal of this paper is the development and analysis of methods that are able to capitalize on such situations, while still being endowed with optimal worst-case guarantees.

## 2.1 Exponentially weighted forecasters

All proposed algorithms in this paper are variations of *exponentially weighted forecasters* [2]. For each time $t \in [n]$, such algorithms assign a weight $w_{i,t} \geq 0$ to the $i$th expert. The forecast prediction at time $t$ and decision whether to observe the outcome or not are randomized, and based exclusively on the expert weights and the expert predictions at that time. Therefore, $\hat{y}_t \sim \text{Ber}(p_t)$ and $Z_t \sim \text{Ber}(q_t)$ are conditionally independent Bernoulli random variables given $p_t$ and $q_t$. Here, $p_t$ and $q_t$ depend on the past only via the weights $\{w_{i,j-1}\}_{i \in [N], j \in [t]}$ and the current expert predictions $\{f_{i,t}\}_{i \in [N]}$. The exact specifications of $p_t$ and $q_t$ depend on the setting and assumptions under consideration.

After a prediction has been made, the weights for all experts are updated using the exponential weights update based on the importance-weighted losses $\ell_{i,t} Z_t / q_t$. Specifically,

$$w_{i,t} = w_{i,t-1} \, e^{-\eta \frac{\ell_{i,t} Z_t}{q_t}} \,, \tag{1}$$

where $\eta > 0$ is the learning rate. To ensure that the definition in (1) is sound for any $q_t \geq 0$, we set $w_{i,t} = w_{i,t-1}$ if $q_t = 0$. Finally, we define the weighted average of experts predicting label 1 at time $t$ as

$$A_{1,t} := \frac{\sum_{i=1}^N w_{i,t-1} f_{i,t}}{\sum_{i=1}^N w_{i,t-1}}. \tag{2}$$

This quantity plays a crucial role in our sampling strategy. We will use the name exponentially weighted forecaster liberally to refer to any forecaster for which $p_t$ is a function of $A_{1,t}$. Throughout, we assume that the weights for all forecasters are uniformly initialized as $w_{i,0} = 1/N$ for $i \in [N]$.

## 3 Regret bounds with a perfect expert

In this section, we consider a very optimistic scenario where one expert is perfect, in the sense that it does not make any mistakes. The results and derivation for this setting are didactic, and pave the way for more general scenarios where this assumption is dropped. We say that the $i$th expert is perfect if $\ell_{i,t} = 0$ for all $t \in [n]$. The existence of such an expert implies that $\min_{i \in [N]} L_{i,n} = 0$. Therefore, the regret of a forecaster is simply the number of errors it makes, that is, $R_n = L_n$. In such a scenario, any reasonable algorithm should immediately discard experts as soon as they make even a single mistake. For an exponentially weighted forecaster, this is equivalent to setting $\eta = \infty$. Due to the uniform weight initialization, the scaled weight vector $N \cdot (w_{1,t}, \ldots, w_{N,t})$ is thus binary, and indicates which experts agree with all the observed outcomes up to time $t$.

First, consider a scenario where the forecaster always collects feedback, that is, $q_t = 1$ for all $t \in [n]$. A natural forecasting strategy at time $t$ is to *follow the majority*, that is, to predict according to the

majority of the experts that have not made a mistake so far. When the forecaster predicts the wrong label, this implies that at least half of the experts still under consideration are not perfect. Since the number of experts under consideration is at least halved for each mistake the forecaster incurs, this strategy is guaranteed to make at most $\log_2 N$ mistakes. Therefore, we have

$$R_n = L_n \leq \log_2 N \ . \tag{3}$$

Clearly, this implies the following bound on the expected cumulative loss, and thus the regret:

$$\bar{L}_n^{(N)} := \mathbb{E}[L_n] \leq \log_2 N \ . \tag{4}$$

Here, the superscript $(N)$ explicitly denotes the dependence on the number of experts. This bound is tight when the minority is always right and nearly equal in size to the majority.

A natural question to ask is if there exists an algorithm that achieves the expected cumulative loss bound (4) while not necessarily collecting all labels. This is, in fact, possible. The most naive approach is to not collect a label if all experts still under consideration agree on their prediction, as in that case, they must all be correct due to the existence of a perfect expert. However, a more refined strategy that can collect fewer labels is possible, leading to the following theorem.

**Theorem 1.** *Consider the exponentially weighted follow-the-majority forecaster with $\eta = \infty$. Specifically, let $p_t = \mathbb{1}\{A_{1,t} \geq 1/2\}$, so that $\hat{y}_t = \mathbb{1}\{A_{1,t} \geq 1/2\}$. Furthermore, let*

$$q_t = \begin{cases} 0 & \text{if } A_{1,t} \in \{0,1\}, \\ -\frac{1}{\log_2 \min(A_{1,t}, 1-A_{1,t})} & \text{otherwise.} \end{cases}$$

*For this forecaster, we have*

$$\bar{L}_n^{(N)} \leq \log_2 N \ .$$

Recall that $A_{1,t}$ is simply the proportion of experts still under consideration that predict $y_t = 1$. It is insightful to look at the expression for $q_t$, as it is somewhat intuitive. The bigger the disagreement between the experts' predictions, the higher the probability that we collect a label. Conversely, when $A_{1,t}$ approaches either 0 or 1, $q_t$ quickly approaches zero, meaning we rarely collect a label. Theorem 1 tells us that, remarkably, we can achieve the same worst-case bound as the full-information forecaster while sometimes collecting much less feedback. The proof of this result, in Appendix A, uses a clean induction argument that constructively gives rise to the expression for $q_t$. This principled way of reasoning identifies, in a sense, the best way to assess disagreement between experts: the specified $q_t$ is the lowest possible sampling probability that preserves worst-case regret guarantees.

A slightly better regret bound is possible by using a variation of follow the majority, called the boosted majority of leaders. For this algorithm, the upper bound is endowed with a matching lower bound (including constant factors). In Appendix B, we devise a label-efficient version of the boosted majority of leaders, retaining the same worst-case regret bound as its full-information counterpart.

## 4  General regret bounds without a perfect expert

In this section, we drop the assumption of the existence of a perfect expert. It is therefore no longer sensible to use an infinite learning rate $\eta$, since this would discard very good experts based on their first observed error. We consider the general exponentially weighted forecaster described in Section 2.1, now simply with $p_t = A_{1,t}$.

For the scenario where $q_t = 1$ for all $t$, a classical regret bound is well-known (see, for instance, [3, Thm 2.2]). Specifically, for the general exponentially weighted forecaster, with $p_t = A_{1,t}$, $q_t = 1$, and uniform weight initialization, we have

$$\bar{R}_n := \mathbb{E}[R_n] = \mathbb{E}\left[L_n - \min_{i \in [N]} L_{i,n}\right] \leq \frac{\ln N}{\eta} + \frac{n\eta}{8} \ . \tag{5}$$

In Theorem 2 below, we prove a stronger version of (5) that allows for an adaptive label-collection procedure. As before, we focus on the expected regret, $\bar{R}_n = \mathbb{E}[R_n]$. As done in Section 3 for the case of a perfect expert, we identify an expression for $q_t$, which is not necessarily identically 1, but still ensures the bound in (5) is valid. To state our main result, we need the following definition, which is guaranteed to be sound by Lemma 1.

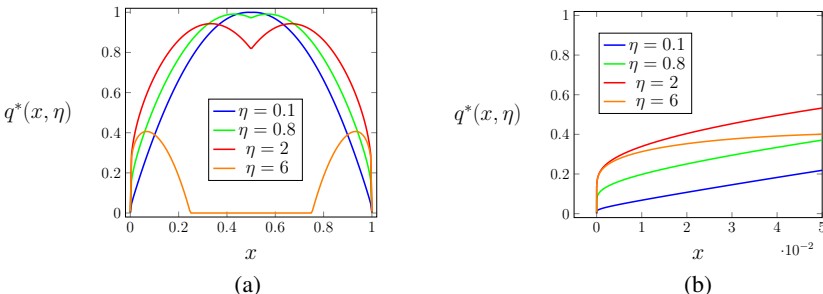

Figure 1: The function $q^*(x, \eta)$ for various values of $\eta$. Panel (b) is a zoomed version of panel (a).

**Definition 1.** *For $x \in [0, 1]$ and $\eta > 0$, define*

$$q^*(x, \eta) = \inf \left\{ q \in (0, 1] : x + \frac{q}{\eta} \ln \left( 1 - x + xe^{-\eta/q} \right) \le \frac{\eta}{8}, \tag{6} \right.$$

$$\left. 1 - x + \frac{q}{\eta} \ln \left( x + (1 - x)e^{-\eta/q} \right) \le \frac{\eta}{8} \right\} .$$

In the following theorem, we present the label-efficient version of (5).

**Theorem 2.** *Consider an exponentially weighted forecaster with $p_t = A_{1,t}$ and*

$$q_t \ge q^*(A_{1,t}, \eta) := q_t^* .$$

*For this forecaster, we have*

$$\bar{R}_n = \mathbb{E} \left[ L_n - \min_{i \in [N]} L_{i,n} \right] \le \frac{\ln N}{\eta} + \frac{n\eta}{8} . \tag{7}$$

The proof, which is deferred to Appendix C, is similar to that used for Theorem 1, but with key modifications to account for the lack of a perfect expert. In particular, we need to account for the finite, importance-weighted weight updates, and carefully select $q_t^*$ accordingly. While the proof allows for non-uniform weight initializations, we focus on the uniform case, as this enables us to optimally tune the learning rate. The result for general weight initializations is given in Appendix C.

Theorem 2 shows that the proposed label-efficient forecaster satisfies the same expected regret bound as the exponentially weighted forecaster with $q_t := 1$. While the expression for $q^*(x, \eta)$ in (6) is somewhat opaque, the underlying motivation is constructive, and it arises naturally in the proof of the theorem. In fact, $q_t^*$ is the smallest possible label-collection probability ensuring the regret bound (7). One may wonder if $q_t^*$ is well defined, as it is the infimum of a set that may be empty. However, as shown in the following lemma, this set always contains the point 1, ensuring that $q_t^* \le 1$.

**Lemma 1.** *For all $\eta > 0$ and $x \in [0, 1]$, we have*

$$1 \in \left\{ q \in (0, 1] : x + \frac{q}{\eta} \ln \left( 1 - x + xe^{-\eta/q} \right) \le \frac{\eta}{8} \right\} .$$

The proof is presented in Appendix D, and is essentially a consequence of Hoeffding's inequality. While $q^*(x, \eta)$ does not admit an analytic solution, its behavior as a function of $x$, depicted in Figure 1, is rather intuitive. Since $\eta = \sqrt{8(\ln N)/n}$ minimizes the regret bound (7), we are primarily interested in small values of $\eta$. When the learning rate $\eta$ is not too large, the behavior of $q_t^*$ can be interpreted as follows: the larger the (weighted) disagreement of the experts is, the closer the value of $A_{1,t}$ is to the point $1/2$. In this case, $q_t^*$ will be close to 1, and we collect a label with high probability. When $A_{1,t}$ is close to 0 or 1, the (weighted) experts essentially agree, so the probability of collecting a label will be small. For large learning rates, the behavior of $q_t^*$ appears a bit strange, but note that for $\eta \ge 8$, the regret bound is vacuous. Thus, for this case, $q^*(x, \eta) = 0$ for all $x \in [0, 1]$.

The regret guarantee in Theorem 2 is valid provided one uses any choice $q_t \ge q_t^*$. The following lemma provides both an asymptotic characterization of $q_t^*$ as $\eta \to 0$, as well as a simple upper bound that can be used for both analytical purposes and practical implementations.

**Lemma 2.** *For any $x \in [0, 1]$, we have*

$$\lim_{\eta \to 0} q^*(x, \eta) = 4x(1 - x) .$$

*Furthermore, for any $\eta > 0$ and $x \in [0, 1]$,*

$$q^*(x, \eta) \leq \min(4x(1 - x) + \eta/3, 1) . \tag{8}$$

The proof of this result is somewhat technical and tedious, and deferred to Appendix E. In the remainder of this paper, we will use this upper bound extensively.

## 5 Label complexity

We now examine the label complexity, defined as $S_n := \sum_{t=1}^n Z_t$. In [10, Thm. 13], it is shown that there exists a setting for which the expected regret of a forecaster that collects $m$ labels is lower-bounded by $cn\sqrt{\ln(N-1)/m}$ for some constant $c$. Hence, in the worst case, the number of collected labels needs to be linear in $n$ in order to achieve an expected regret that scales at most as $\sqrt{n}$. However, since $q_t^*$ can be less than 1, it is clear that the label-efficient exponentially weighted forecaster from Theorem 2 can collect fewer than $n$ labels in more benign settings. To this end, we consider a scenario with a unique best expert, which at each round is separated from the rest in terms of its expected loss. To state the condition precisely, we need to define $\mathbb{E}_t = \mathbb{E}[ \cdot \mid \mathcal{F}_{t-1}]$ as the expectation at time $t$ conditional on all possible randomness up to time $t - 1$, that is, for $\mathcal{F}_t = \sigma(\{Z_j, y_j, f_{1,j}, \ldots, f_{N,j}\}_{j=1,\ldots,t})$. With this, we assume that there is a unique expert $i^* \in [N]$ such that, for all $i \neq i^*$ and $t \in [n]$,

$$\mathbb{E}_t[\ell_{i,t} - \ell_{i^*,t}] \geq \Delta > 0 \qquad \text{almost surely.}$$

The parameter $\Delta$ characterizes the difficulty of the given learning problem. If $\Delta$ is large, the best expert significantly outperforms the others, and is thus easily discernible, whereas if $\Delta$ is small, the best expert is harder to identify. In particular, if the vectors $(y_t, f_{1,t}, \ldots, f_{N,t})$ are independent and identically distributed over rounds $t \in [n]$, $\Delta$ is just the difference in expected loss between the best and the second-best expert in a single round, which is a common measure of difficulty for stochastic bandits [33, Thm. 2.1]. This difficulty measure has also been used in the context of prediction with expert advice [34]. Similar stochastic assumptions are standard in (batch) active learning, and highly relevant in practical settings (see [14, 15, 18, 21, 27] and references therein). Strictly speaking, our result holds under a more general assumption, where the best expert *emerges after a time $\tau^*$* instead of being apparent from the first round. This means that the best expert is even allowed to perform the worst for some rounds, as long as it performs well in sufficiently many other rounds. While we state and prove the result under this more general condition in Appendix F, we present the simpler assumption here for clarity.

We now state our main result for the label complexity.

**Theorem 3.** *Consider the label-efficient exponentially weighted forecaster from Theorem 2 with $q_t = \min(4A_{1,t}(1 - A_{1,t}) + \eta/3, 1)$ and any $\eta > 0$. Suppose that there exists a single best expert $i^*$ such that, for all $i \neq i^*$ and all $t \in [n]$,*

$$\mathbb{E}_t[\ell_{i,t} - \ell_{i^*,t}] \geq \Delta > 0 \qquad \text{almost surely.}$$

*Then, for any $n \geq 4$, the expected label complexity is at most*

$$\mathbb{E}[S_n] \leq \frac{50}{\eta \Delta^2} \ln \left( \frac{N \ln n}{\eta} \right) + 3\eta n + 1 . \tag{9}$$

*Proof sketch.* Initially, the sampling probability $q_t$ is large, but as we collect more labels, it will become detectable that one of the experts is better than the others. As this happens, $q_t$ will tend to decrease until it (nearly) reaches its minimum value $\eta/3$. We therefore divide the forecasting process into time $t \leq \tau$ and $t > \tau$. With a suitable choice of $\tau \approx 1/(\eta \Delta^2)$ (up to logarithmic factors), we can guarantee that the sampling probability is at most $q_t \leq 4\eta/3$ for all $t > \tau$ with sufficiently high probability. This is shown by controlling the deviations of the cumulative importance-weighted loss differences $\tilde{\Lambda}_t^i = \sum_{j=1}^t (l_{i,j} - l_{i^*,j})/q_j$ for $i \neq i^*$ from their expected values by using an anytime version of Freedman's inequality. With this, we simply upper bound the label complexity for the first $\tau$ rounds by $\tau$, and over the remaining rounds, the expected number of collected labels is roughly $\eta(n - \tau) \leq \eta n$. This leads to a total expected label complexity of $1/(\eta \Delta^2) + \eta n$, up to logarithmic factors. The full proof is deferred to Appendix F. □

As mentioned earlier, the learning rate optimizing the regret bound (7) is $\eta = \sqrt{8\ln(N)/n}$. For this particular choice, the label complexity in (9) is roughly $\sqrt{n}/\Delta^2$, up to constants and logarithmic factors. We have thus established that the label-efficient forecaster achieves the best of both worlds: it queries sufficiently many labels in the worst case to obtain optimal regret guarantees, while simultaneously querying much fewer labels in more benign settings. It is interesting to note that the label complexity dependence of $1/\Delta^2$ on $\Delta$ is less benign than the dependence of the regret bound from, e.g., [34, Thm. 11], which is $1/\Delta$. The underlying reason for this is that, while the two are similar, the label complexity is not directly comparable to the regret. In particular, the label complexity has much higher variance.

The bound of Theorem 3 relies on setting the sampling probability $q_t$ to be the upper bound on $q_t^*$ from Lemma 2. This bound is clearly loose when $q_t^*$ is approximately zero, and one may wonder if the label complexity of the algorithm would be radically smaller when using a forecaster for which $q_t = q_t^*$ instead. With the choice $\eta = \sqrt{8\ln(N)/n}$, which optimizes the bound in (7), it seems unlikely that the label complexity will substantially change, as numerical experiments suggest that the label complexity attained with $q_t$ set as $q_t^*$ or the corresponding upper bound from (8) appear to be within a constant factor. That being said, for larger values of $\eta$, the impact of using the upper bound in (8) is likely much more dramatic.

## 6  Numerical experiments

To further assess the behavior of the label-efficient forecaster from Theorem 2, we consider a classical active learning scenario in a batch setting, for which there are known minimax rates for the risk under both active and passive learning paradigms. We will set the sampling probability to be

$$q_t = \min(4A_{1,t}(1 - A_{1,t}) + \eta/3, 1) \ .$$

Let $D_n = ((X_t, Y_t))_{t=1}^n$ be an ordered sequence of independent and identically distributed pairs of random variables with joint distribution $D$. The first entry of $(X_i, Y_i)$ represents a feature, and the second entry is the corresponding label. The goal is to predict the label $Y_i \in \{0, 1\}$ based on the feature $X_i$. Specifically, we want to identify a map $(x, D_n) \mapsto \hat{g}_n(x, D_n) \in \{0, 1\}$ such that, for a pair $(X, Y) \sim D$ that is drawn independently from $D_n$, we have small (zero-one loss) expected risk

$$\mathrm{Risk}(\hat{g}_n) := \mathbb{P}(\hat{g}_n(X, D_n) \neq Y) \ .$$

Concretely, we consider the following scenario, inspired by the results in [14]. Let the features $X_i$ be uniformly distributed in $[0, 1]$, and $Y_i \in \{0, 1\}$ be such that $\mathbb{P}(Y_i = 1 | X_i = x) = \zeta(x)$. Specifically, let $\tau_0 \in [0, 1]$ such that $\zeta(x) \geq 1/2$ when $x \geq \tau_0$ and $\zeta(x) \leq 1/2$ otherwise. Furthermore, assume that for all $x \in [0, 1]$, $\zeta(x)$ satisfies

$$c|x - \tau_0|^{\kappa-1} \leq |\zeta(x) - 1/2| \leq C|x - \tau_0|^{\kappa-1} \ ,$$

for some $c, C > 0$ and $\kappa > 1$. If $\tau_0$ is known, the optimal classifier is simply $g^*(x) = \mathbb{1}\{x \geq \tau_0\}$. The minimum achievable excess risk when learning $\hat{g}_n$ from $D_n$ in this type of problems has been studied in, e.g., [14, 35]. For this setting, it is known that

$$\inf_{\hat{g}_n} \sup_{\tau_0 \in [0,1]} \mathrm{Risk}(\hat{g}_n) - \mathrm{Risk}(g^*) \asymp n^{\frac{-\kappa}{2\kappa-1}} \ ,$$

as $n \to \infty$. However, rather than the classical supervised learning setting above, we can instead consider active learning procedures. Specifically, consider a sequential learner that can generate feature-queries $X_i'$ and sample a corresponding label $Y_i'$, such that $\mathbb{P}(Y_i' = 1 | X_i' = x) = \zeta(x)$. This is often referred to as pool-based active learning. At time $t$, the learner can choose $X_t'$ as a function of the past $((X_j', Y_j'))_{j=1}^{t-1}$ according to a (possibly random) sampling strategy $\mathcal{A}_n$. This extra flexibility allows the learner to carefully select informative examples to guide the learning process. Similarly to the passive learning setting, the ultimate goal is to identify a prediction rule $(x, D_n') \mapsto \hat{g}_n^A(x, D_n') \in \{0, 1\}$, where $D_n' = ((X_t', Y_t'))_{t=1}^n$. In [14], it is shown that for this active learning setting, the minimax rates are also known, and given by

$$\inf_{\hat{g}_n^A, \mathcal{A}_n} \sup_{\tau_0 \in [0,1]} \mathrm{Risk}(\hat{g}_n^A) - \mathrm{Risk}(g^*) \asymp n^{\frac{-\kappa}{2\kappa-2}} \ ,$$

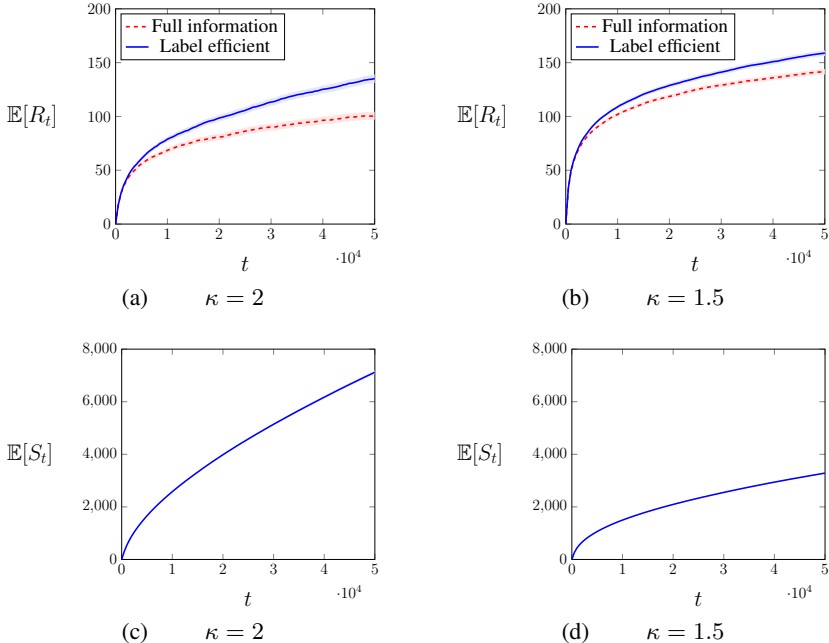

Figure 2: Numerical results for expected regret and label complexity when $n = 50000$ and $N = 225$. Panels (a) and (b) depict the expected regret $\mathbb{E}[R_t]$ as a function of $t$, for $\kappa = 2$ and $\kappa = 1.5$ respectively. Panels (c) and (d) depict the expected label complexity of the label-efficient forecaster, $\mathbb{E}[S_t]$, as a function of $t$ for $\kappa = 2$ and $\kappa = 1.5$ respectively. The expectations were estimated from 500 independent realizations of the process and the shaded areas indicate the corresponding pointwise 95% confidence intervals.

as $n \to \infty$. This shows that there are potentially massive gains for active learning, particularly when $\kappa$ is close to 1. A natural question is whether similar conclusions hold for streaming active learning. In this setting, instead of selecting which example $X_i'$ to query, the learner observes the features $(X_1, \ldots, X_n)$ sequentially, and decides at each time $t$ whether or not it should query the corresponding label. This is analogous to the online prediction setting discussed in this paper.

We now study this setting numerically. For the simulations, we use the specific choice

$$\zeta(x) = \frac{1}{2} + \frac{1}{2}\mathrm{sign}(x - \tau_0)|x - \tau_0|^{\kappa-1} \ ,$$

to generate sequences $(Y_1, \ldots, Y_n)$, based on a sequence of features $(X_1, \ldots, X_n)$ sampled from the uniform distribution on $[0, 1]$. Furthermore, we consider the class of $N$ experts such that

$$f_{i,t} = \mathbb{1}\left\{X_t \geq \frac{i-1}{N-1}\right\} \ ,$$

with $i \in [N]$ and $t \in [n]$.

## 6.1 Expected regret and label complexity

In the simulations, we set $\tau_0 = 1/2$ and $N = \lceil\sqrt{n}\rceil + \mathbb{1}\{\lceil\sqrt{n}\rceil \text{ is even}\}$. This choice enforces that $N$ is odd, ensuring the optimal classifier is one of the experts. Throughout, we set $\eta = \sqrt{8\ln(N)/n}$, which minimizes the regret bound (7). First, we investigate the expected regret relative to the optimal prediction rule for the label-efficient exponentially weighted forecaster with $q_t$ given by (8), and compare it with the corresponding regret for the full-information forecaster that collects all labels. Specifically, the regret at time $t$ of a forecaster that predicts $\{\hat{Y}_j\}_{j=1}^n$ is given by

$$\mathbb{E}[R_t] = \sum_{j=1}^{t} \mathbb{E}[\ell(\hat{Y}_t, Y_t) - \ell(g^*(X_t), Y_t)] \ .$$

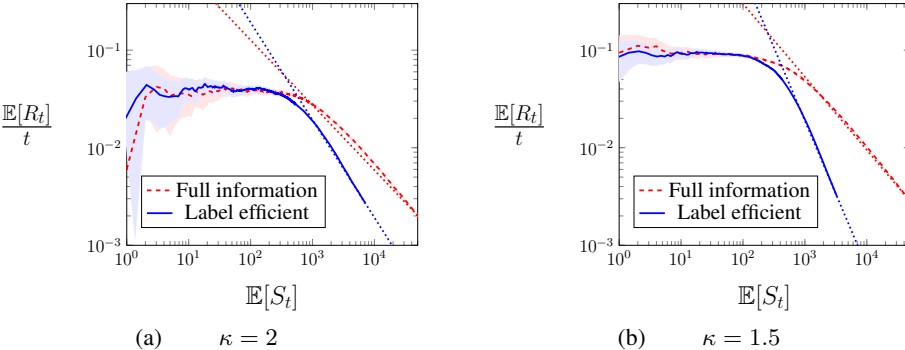

Figure 3: Numerical results for the normalized regret as a function of the expected label complexity when $n = 50000$ and $N = 225$. The straight dotted lines are displayed for comparison, and have slopes given by $-\kappa/(2\kappa - 1)$ (full information) and $-\kappa/(2\kappa - 2)$ (label efficient). The expectations were estimated from $500$ independent realizations of the process and the shaded areas indicate the corresponding pointwise $95\%$ confidence intervals.

Furthermore, to study the potential reduction in the number of collected labels, we also evaluate the expected label complexity $\mathbb{E}[S_t]$ of the label-efficient forecaster. To approximate the expectations above, we use Monte-Carlo averaging with $500$ independent realizations. Further experimental details are given in Appendix G. The results are shown in Figure 2. We see that the regret is comparable for the full-information and label-efficient forecasters, albeit slightly higher for the latter. Since $P(g^*(X) \neq Y) = \frac{1}{2} - \frac{1}{\kappa 2^\kappa}$, the expected cumulative loss of the optimal classifier grows linearly with $t$. For instance, when $\kappa = 2$, we have $\sum_{j=1}^t \mathbb{E}[\ell(g^*(X_t), Y_t)] = 3t/8$. Hence, the regret relative to the best expert is much smaller than the pessimistic (i.e., worst-case for adversarial environments) bound in (7). We also observe that the expected label complexity grows sub-linearly with $t$, as expected, and that $\mathbb{E}[S_n] \ll n$, demonstrating that a good prediction rule can be learned with relatively few labels. When $\kappa = 1.5$, the number of collected labels is significantly smaller than when $\kappa = 2$. This is in line with the known minimax rates for active learning from [14]. To further examine this connection, we now turn to normalized regret.

## 6.2 Normalized regret relative to the number of samples

To relate the results of the label-efficient forecaster with known minimax rates for active learning, we investigate the expected regret normalized by the number of samples. Specifically, let

$$r(t) = \frac{1}{t} \mathbb{E}[R_t] = \frac{1}{t} \sum_{j=1}^t \mathbb{E}[\ell(\hat{Y}_t, Y_t) - \ell(g^*(X_t), Y_t)] \,.$$

For the full-information forecaster, we expect that $r(t) \asymp t^{-\kappa/(2\kappa-1)}$ as $t \to \infty$. The same holds for the label-efficient forecaster, but in this case, the relation between $r(t)$ and the expected number of collected labels $\mathbb{E}[S_t]$ is more interesting. If the label-efficient forecaster performs optimally, we expect $r(t) \asymp \mathbb{E}[S_t]^{-\kappa/(2\kappa-2)}$ as $t \to \infty$. To examine this, we plot $r(t)$ against $\mathbb{E}[S_t]$ (which equals $t$ for the full-information forecaster) in logarithmic scales, so the expected asymptotic behavior corresponds to a linear decay with slopes given by $-\kappa/(2\kappa - 1)$ for the full-information forecaster and $-\kappa/(2\kappa - 2)$ for the label-efficient forecaster. This is shown in Figure 3 for $\kappa = 1.5$ and $\kappa = 2$.

We see that the observed behavior is compatible with the known asymptotics for active learning, and similar results arise when considering different values of $\kappa$. More importantly, it appears that the label-efficient forecaster optimally adapts to the underlying setting. This is remarkable, as the label-efficient forecaster does not rely on any domain knowledge. Indeed, it has no knowledge of the statistical setting, and in particular, it has no knowledge of the parameter $\kappa$, which encapsulates the difficulty of the learning task. Note that our regret bounds are too loose to provide a theoretical justification of these observations via an online-to-batch conversion, and that such theoretical analyses will only be fruitful when considering non-parametric classes of experts, for which the asymptotics of the excess risk are $\omega(1/\sqrt{n})$.

## 7 Discussion and outlook

In this paper, we presented a set of adaptive label-efficient algorithms. These follow from a very straightforward design principle, namely, identifying the smallest possible label collection probability $q_t$ that ensures that a known worst-case expected regret bound is satisfied. This leads to simple, yet powerful, algorithms, endowed with best-of-both-worlds guarantees. We conjecture that a similar approach can be used for a broader class of prediction tasks and losses than what is considered in this paper. For instance, the results we present can be straightforwardly extended to a setting where the expert outputs take values in $[0, 1]$, as long as the label sequence and forecaster prediction remain binary and take values in $\{0, 1\}$. In fact, the same inductive approach can be used when $y_t \in [0, 1]$ and one considers a general loss function. However, the resulting label collection probability will be significantly more complicated than that of Definition 1. An interesting side effect of our analysis is that it leads to an inductive proof of the regret bound for standard, full-information algorithms. Extending the label complexity result, and in particular connecting it with known minimax theory of active learning in statistical settings, remains an interesting avenue for future research. Finally, another intriguing direction is to extend our approach to the bandit setting. In the setting we consider in this paper, we observe the losses of all experts when observing a label. In contrast, in the bandit setting, only the loss of the selected arm would be observed for each round. This would necessitate the forecaster to incorporate more exploration in its strategy, and the analysis of a label-efficient version seems like it would be quite different from what is used in this paper, although some of the ideas may transfer.

### Acknowledgements

The authors would like to thank Wojciech Kotłowski, Gábor Lugosi, and Menno van Eersel for fruitful discussions that contributed to this work. The algorithmic ideas underlying this work were developed when R. Castro was a research fellow at the University of Wisconsin – Madison. This work was partly done while F. Hellström was visiting the Eindhoven University of Technology and the University of Amsterdam supported by EURANDOM and a STAR visitor grant, the Wallenberg AI, Autonomous Systems and Software Program (WASP) funded by the Knut and Alice Wallenberg Foundation, and the Chalmers AI Research Center (CHAIR). T. van Erven was supported by the Netherlands Organization for Scientific Research (NWO), grant number VI.Vidi.192.095.

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

# A Proof of Theorem 1

**Theorem 1.** *Consider the exponentially weighted follow the majority forecaster with $\eta = \infty$. Specifically, let $p_t = \mathbb{1}\{A_{1,t} \geq 1/2\}$, so that $\hat{y}_t = \mathbb{1}\{A_{1,t} \geq 1/2\}$. Furthermore, let*

$$q_t = \begin{cases} 0 & \text{if } A_{1,t} \in \{0,1\}, \\ -\frac{1}{\log_2 \min(A_{1,t}, 1-A_{1,t})} & \text{otherwise.} \end{cases}$$

*For this forecaster, we have*

$$\bar{L}_n^{(N)} \leq \log_2 N .$$

*Proof.* The main idea is to proceed by induction on $n$. For $n = 1$, the result holds trivially, regardless of the choice for $q_t$. Now, suppose that $\bar{L}_t^{(N)} \leq \log_2 N$ for all values $t \in [n-1]$, any sequence of observations and expert predictions, and any number of experts $N$. Based on this assumption, we will derive a bound for $\bar{L}_n^{(N)}$. Let $k := \sum_{i=1}^N \ell_{i,1}$ be the number of experts that make a mistake when $t = 1$. Note that $0 \leq k \leq N - 1$, as there is one perfect expert. When $k < N/2$, the majority vote $\hat{y}_1$ is necessarily equal to $y_1$. Therefore,

$$\begin{aligned} \bar{L}_n^{(N)} &= \mathbb{E}[\ell(\hat{y}_1, y_1)] + \sum_{t=2}^n \mathbb{E}[\ell(\hat{y}_t, y_t)] = \sum_{t=2}^n \mathbb{E}[\ell(\hat{y}_t, y_t)] \\ &= \sum_{t=2}^n q_1 \mathbb{E}[\ell(\hat{y}_t, y_t)|Z_1 = 1] + (1 - q_1)\mathbb{E}[\ell(\hat{y}_t, y_t)|Z_1 = 0] \\ &= q_1 \bar{L}_{2:n}^{(N-k)} + (1 - q_1)\bar{L}_{2:n}^{(N)} \\ &\leq \log_2 N , \end{aligned} \tag{10}$$

where $\bar{L}_{2:n}^{(N)}$ denotes the expected cumulative loss in rounds $2, \ldots, n$ with $N$ experts. Because the internal state of the algorithm only consists of a list of experts that have made no errors so far, the task in rounds $2, \ldots, n$ is equivalent to a task over $n - 1$ rounds starting with the experts that remain after round 1. Therefore, we can apply the induction hypothesis to obtain $\bar{L}_{2:n}^{(N-k)} \leq \log_2(N-k) \leq \log_2 N$ and $\bar{L}_{2:n}^{(N)} \leq \log_2 N$, which justifies the last inequality. We conclude that, when $k < N/2$, the bound holds regardless of the choice of $q_1$.

On the other hand, if $N/2 \leq k \leq N - 1$, the forecaster incurs an error at time $t = 1$. Using the induction hypothesis and an analogous reasoning as above, we find that

$$\bar{L}_n^{(N)} = 1 + q_1 \bar{L}_{2:n}^{(N-k)} + (1 - q_1)\bar{L}_{2:n}^{(N)} \leq 1 + q_1 \log_2(N - k) + (1 - q_1)\log_2(N) . \tag{11}$$

To ensure that the right-hand-side of (11) satisfies the desired bound, we need to select $q_1$ such that

$$1 + q_1 \log_2(N - k) + (1 - q_1)\log_2(N) \leq \log_2 N ,$$

which can be written equivalently as

$$\frac{1}{q_1} \leq \log_2 N - \log_2(N - k) .$$

If we do not observe a label, we do not know the value $k$. All we know is that, since $y_1 \in \{0, 1\}$, we have $k \in \left\{\sum_{i=1}^N f_{i,1}, N - \sum_{i=1}^N f_{i,1}\right\}$. Therefore, to ensure the induction proof works, we need

$$q_1 \geq -\frac{1}{\log_2 \min(A_{1,1}, 1 - A_{1,1})} ,$$

where $A_{1,1} = \frac{1}{N}\sum_{i=1}^N f_{i,1}$. Note that the case $k = N$ cannot occur as there is always a perfect expert, but to ensure that the above definition is sound, we define $q_1 = 0$ when $A_{1,1} \in \{0, 1\}$. $\quad\square$

# B Label-efficient boosted majority of leaders

Through a variation of follow the majority, a slightly better regret bound than the one in Theorem 1 can be obtained, as well as a matching lower bound. As shown by [36, Proposition 18.1.3], there exists an adversarial strategy for the environment such that any forecaster will incur at least $\lfloor \log_2 N \rfloor / 2 \geq \lfloor \log_4 N \rfloor$ errors in expectation. A matching upper bound can be obtained, when $N$ is a power of two, by considering a forecaster that incorporates randomness in its predictions. This is referred to as the *boosted majority of leaders*. As shown in the following theorem, this procedure can be made label-efficient while ensuring the same expected regret bound.

**Theorem 4.** *Consider an exponentially weighted forecaster for which $\eta = \infty$ and*

$$p_t = \begin{cases} 0 & \text{if } A_{1,t} \leq 1/4 \\ 1 + \log_4 A_{1,t} & \text{if } 1/4 < A_{1,t} \leq 1/2 \\ -\log_4(1 - A_{1,t}) & \text{if } 1/2 < A_{1,t} \leq 3/4 \\ 1 & \text{if } A_{1,t} > 3/4 \end{cases}$$

*and*

$$q_t = \begin{cases} 0 & \text{if } A_{1,t} = 0 \\ -1/\log_4 A_{1,1} & \text{if } 0 < A_{1,t} < 1/4 \\ 1 & \text{if } 1/4 \leq A_{1,t} \leq 3/4 \\ -1/\log_4(1 - A_{1,1}) & \text{if } 3/4 < A_{1,t} < 1 \\ 0 & \text{if } A_{1,t} = 1 \end{cases} .$$

*For this forecaster, we have*

$$\bar{L}_n^{(N)} \leq \log_4 N .$$

*Proof.* The proof technique is analogous to that of Theorem 1: we use an induction argument to find a choice of $q_t$ that guarantees the desired regret bound. We will proceed by analyzing different cases, depending on the value of $A_{1,t}$. First, assume that $A_{1,1} \leq 1/4$. If $y_1 = 0$, then

$$\bar{L}_n^{(N)} = 0 + q_1 \bar{L}_{2:n}^{(N(1-A_{1,1}))} + (1 - q_1)\bar{L}_{2:n}^{(N)} .$$

Thus, taking $q_1 \geq 0$ suffices. If $y_1 = 1$, then

$$\bar{L}_n^{(N)} = 1 + q_1 \bar{L}_{2:n}^{(NA_{1,1})} + (1 - q_1)\bar{L}_{2:n}^{(N)} ,$$

and, using the same reasoning as before, it suffices to take

$$q_1 \geq -\frac{1}{\log_4 A_{1,1}} .$$

An analogous reasoning applies when $A_{1,1} > 3/4$, so that for this case, it suffices to take

$$q_1 \geq -\frac{1}{\log_4(1 - A_{1,1})} .$$

Now, consider the case $1/4 < A_{1,1} \leq 1/2$. If $y_1 = 0$, then

$$\bar{L}_n^{(N)} = 1 + \log_4 A_{1,1} + q_1 \bar{L}_{2:n}^{(N(1-A_{1,1}))} + (1 - q_1)\bar{L}_{2:n}^{(N)}$$
$$\leq 1 + \log_4 A_{1,1} + q_1 \log_4(N(1 - A_{1,1})) + (1 - q_1)\log_4 N ,$$

which implies that

$$q_1 \geq -\frac{1 + \log_4 A_{1,1}}{\log_4(1 - A_{1,1})} .$$

Similarly, if $y_1 = 1$, we must have

$$q_1 \geq -\frac{-\log_4 A_{1,1}}{\log_4 A_{1,1}} = 1 .$$

Since we do not know $y_1$ before the decision, the only possibility is to take $q_1 = 1$.

A similar reasoning applies to the case $1/2 < A_{1,1} \leq 3/4$. Therefore, the above relations determine the expression for $q_t$ in the theorem, while enforcing the desired regret bound.

$\square$

## C  Proof of Theorem 2

As mentioned after Theorem 2, an analogous result holds for non-uniform weight initializations. We will state and prove this more general result below, from which Theorem 2 as stated in the main text follows as a special case.

**Theorem 2** (with non-uniform weight initialization). *Consider an exponentially weighted forecaster with initial weight vector $\boldsymbol{w}_{\cdot,0} = (w_{1,0}, \ldots, w_{N,0})$ such that $\sum_{i \in [N]} w_{i,0} = 1$, $p_t = A_{1,t}$ and*

$$q_t \geq q^*(A_{1,t}, \eta) := q_t^* \,.$$

*For this forecaster, we have*

$$\mathbb{E}\Big[L_n^{(\boldsymbol{w}_{\cdot,0})}\Big] \leq \mathbb{E}\left[\min_{i \in [N]} \left(L_{i,n} - \frac{\ln w_{i,0}}{\eta}\right)\right] + \frac{n\eta}{8} \,. \tag{12}$$

*where the superscript in $L_n^{(\boldsymbol{w}_{\cdot,0})} := \sum_{t=1}^n \ell(\hat{y}_t, y_t)$ makes the dependence on the initial weights explicit. In particular, for the choice of initial weights $w_{i,0} := 1/N$ for all $i \in [N]$, we have*

$$\bar{R}_n = \mathbb{E}[L_n] - \mathbb{E}\left[\min_{i \in [N]} L_{i,n}\right] \leq \frac{\ln N}{\eta} + \frac{n\eta}{8} \,.$$

*Proof.* The proof strategy is similar to that used in Theorem 1. We begin by noting that at time $t$, the internal state of the label-efficient exponentially weighted forecaster is determined by the weight vector $\boldsymbol{w}_{\cdot,t-1} = (w_{1,t-1}, \ldots, w_{N,t-1})$. Therefore, it suffices to focus on the requirements for $q_1$ for an arbitrary weight vector. As in Theorem 1, we proceed by induction on $n$.

For $n = 1$, the theorem statement is trivially true, as this algorithm coincides with the ordinary exponentially weighted forecaster (also, the right-hand-side of (12) is bounded from below by $1/2$). Proceeding by induction in $n$, suppose (12) holds for $1, \ldots, n-1$ outcomes. Let $\bar{L}_{n|1}^{(\boldsymbol{w}_{\cdot,0})}$ denote the expected cumulative loss of the forecaster given $(y_1, \{f_{i,1}\}_{i=1}^N)$, i.e., the true label and the expert predictions for time $t = 1$:

$$\bar{L}_{n|1}^{(\boldsymbol{w}_{\cdot,0})} = \mathbb{E}\Big[L_n^{(\boldsymbol{w}_{\cdot,0})} \mid y_1, \{f_{i,1}\}_{i=1}^N\Big] \,.$$

Then, we have

$$\bar{L}_{n|1}^{(\boldsymbol{w}_{\cdot,0})} = \mathbb{E}\big[\ell(\hat{y}_1, y_1) \mid y_1, \{f_{i,1}\}_{i=1}^N\big] + \mathbb{E}\left[\sum_{t=2}^n \ell(\hat{y}_t, y_t) \mid y_1, \{f_{i,1}\}_{i=1}^N\right]$$

$$= A_{1,1} + (1 - 2A_{1,1})y_1 + q_1 \mathbb{E}\left[\sum_{t=2}^n \ell(\hat{y}_t, y_t) \mid Z_1 = 1, y_1, \{f_{i,1}\}_{i=1}^N\right]$$

$$+ (1 - q_1)\mathbb{E}\left[\sum_{t=2}^n \ell(\hat{y}_t, y_t) \mid Z_1 = 0, y_1, \{f_{i,1}\}_{i=1}^N\right]$$

$$\leq A_{1,1} + (1 - 2A_{1,1})y_1 + q_1 \bar{L}_{n-1}^{(\boldsymbol{w}_{\cdot,1})} + (1 - q_1)\bar{L}_{n-1}^{(\boldsymbol{w}_{\cdot,0})} \,.$$

In order to prove the desired result, it is enough to show that

$$\bar{L}_{n|1}^{(\boldsymbol{w}_{\cdot,0})} \leq \mathbb{E}\left[\min_{i \in [N]} \left(\ell_{i,1} + L_{i,2:n} - \frac{\ln w_{i,0}}{\eta}\right)\right] + \frac{n\eta}{8} = \mathbb{E}\left[\left(\ell_{i',1} + L_{i',2:n} - \frac{\ln w_{i',0}}{\eta}\right)\right] + \frac{n\eta}{8} \,.$$

Here, we let $L_{i,2:n} := \sum_{t=2}^n \ell_{i,t}$ and let $i'$ denote the $\arg\min$ of the right-hand side. Now, using the induction hypothesis, we obtain

$$\bar{L}_{n|1}^{(\boldsymbol{w}_{\cdot,0})} \leq A_{1,1} + (1 - 2A_{1,1})y_1 + \frac{(n-1)\eta}{8}$$

$$+ \mathbb{E}\left[\min_{i \in [N]} \left(L_{i,2:n} + q_1 \frac{-\ln w_{i,1}}{\eta} + (1 - q_1)\frac{-\ln w_{i,0}}{\eta}\right)\right]$$

$$\leq A_{1,1} + (1 - 2A_{1,1})y_1 + \frac{(n-1)\eta}{8}$$

$$+ \mathbb{E}\left[\left(L_{i',2:n} + q_1 \frac{-\ln w_{i',1}}{\eta} + (1 - q_1)\frac{-\ln w_{i',0}}{\eta}\right)\right] \,.$$

In the last step, we used the fact that since the upper bound holds for the minimum $i$, it holds for $i'$ in particular. To ensure that the bound in the theorem holds, it is thus sufficient to select $A_{1,1}$ such that it satisfies

$$A_{1,1} + (1 - 2A_{1,1})y_1 + \left( \frac{q_1}{\eta} \left( \ln w_{i',0} - \ln w_{i',1} \right) - \ell_{i',1} \right) \leq \frac{\eta}{8} \,.$$

Notice that, after the first observation, we are back in a situation similar to that at time $t = 1$, but possibly with a different weight vector. Specifically, for $i \in [N]$,

$$w_{i,1} = \frac{w_{i,0}e^{-\eta\ell_{i,1}/q_1}}{\sum_{i=1}^{N} w_{i,0}e^{-\eta\ell_{i,1}/q_1}} \,.$$

It is important at this point that $w_{i,1}$ depends on the choice $q_1$, so we cannot simply solve the above equation for $q_1$. To proceed, it is easier to consider the two possible values of $y_1$ separately.

**Case $y_1 = 0$:**   Note that

$$w_{i,1} = \frac{w_{i,0}e^{-\eta\ell(f_{i,1},y_1)/q_1}}{1 - A_{1,1} + A_{1,1}e^{-\eta/q_1}} \,.$$

Therefore, it suffices to have

$$A_{1,1} + \frac{q_1}{\eta} \ln \left( 1 - A_{1,1} + A_{1,1}e^{-\eta/q_1} \right) \leq \frac{\eta}{8} \,. \tag{13}$$

**Case $y_1 = 1$:**   Similarly as above,

$$w_{i,1} = \frac{w_{i,0}e^{-\eta\ell(f_{i,1},y_1)/q_1}}{A_{1,1} + (1 - A_{1,1})e^{-\eta/q_1}} \,.$$

Therefore, it suffices to have

$$1 - A_{1,1} + \frac{q_1}{\eta} \ln \left( A_{1,1} + (1 - A_{1,1})e^{-\eta/q_1} \right) \leq \frac{\eta}{8} \,. \tag{14}$$

As we do not know the values of $y_1$ when computing $q_1$, we must simultaneously satisfy (13) and (14). Nevertheless, these two conditions involve only $\eta$ and $A_{1,1}$. Thus, we can identify the range of values that $q_1$ can take, as a function of $\eta$ and $A_{1,1}$, while still ensuring the desired regret bound. Specifically, we require that $q_1 \geq q_1^*$, where

$$q_1^* := q_1^*(A_{1,1}, \eta) = \inf \left\{ q \in (0,1] : A_{1,1} + \frac{q}{\eta} \ln \left( 1 - A_{1,1} + A_{1,1}e^{-\eta/q} \right) \leq \frac{\eta}{8}, \right.$$

$$\left. 1 - A_{1,1} + \frac{q}{\eta} \ln \left( A_{1,1} + (1 - A_{1,1})e^{-\eta/q} \right) \leq \frac{\eta}{8} \right\} \,.$$

At this point, it might be unclear if $q_1^*$ is well defined, namely, if there always exists $q \in [0,1]$ satisfying both (13) and (14). This is indeed the case, as shown in Lemma 1. By noting that $\mathbb{E}[\bar{L}_{i,n|1}^{(\boldsymbol{w}_{\cdot,0})}] = \bar{L}_{i,n}^{(\boldsymbol{w}_{\cdot,0})}$, we have completed the induction step.

As stated at the beginning of the proof, looking at $q_1$ suffices to determine the general requirements for $q_t$, concluding the proof. The statement given in (7) follows from instantiating the general result with uniform initial weights. $\qquad\square$

## D   Proof of Lemma 1

**Lemma 1.** *For all $\eta > 0$ and $x \in [0,1]$, we have*

$$1 \in \left\{ q \in (0,1] : x + \frac{q}{\eta} \ln \left( 1 - x + xe^{-\eta/q} \right) \leq \frac{\eta}{8} \right\} \,.$$

*Proof.* Let $B \sim \text{Ber}(x)$, with $x \in [0,1]$. Note that $\mathbb{E}[e^{-\eta B}] = (1 - x) + xe^{-\eta}$. Note also that, by [3, Lem. A.1] we have $\ln \mathbb{E}[e^{-\eta B}] \leq -\eta x + \eta^2/8$. Putting these two facts together we conclude that

$$x + \frac{1}{\eta} \ln \left( 1 - x + xe^{-\eta} \right) \leq x - x + \eta/8 = \eta/8 \,.$$

Therefore, the point 1 is always contained in Definition 1, concluding the proof. $\qquad\square$

# E   Proof of Lemma 2

**Lemma 2.** *For any $x \in [0, 1]$, we have*
$$\lim_{\eta \to 0} q^*(x, \eta) = 4x(1 - x) .$$
*Furthermore, for any $\eta > 0$ and $x \in [0, 1]$,*
$$q^*(x, \eta) \leq \min(4x(1 - x) + \eta/3, 1) .$$

*Proof.* As already shown in Lemma 1, we know that $q^*(x, \eta) \leq 1$. Let $\eta > 0$ be arbitrary. Note that $q^*(x, \eta)$ needs to be a solution in $q$ of the following equation:
$$\underbrace{\eta x + q \ln(1 - x + xe^{-\eta/q})}_{:=g(\eta)} \leq \frac{\eta^2}{8} . \tag{15}$$

Note that $\lim_{\eta \to 0} g(\eta) = 0$, so we can extend the definition of $g$ to $0$ by continuity. Specifically,
$$g(\eta) := \begin{cases} \eta x + q \ln(1 - x + xe^{-\eta/q}) & \text{if } \eta > 0 \\ 0 & \text{if } \eta = 0 \end{cases} .$$

We proceed by using a Taylor expansion of $g(\eta)$ around $0$. Tedious, but straightforward computations yields
$$g'(\eta) = \frac{\partial}{\partial \eta} g(\eta) = x \left(1 - \frac{1}{x + (1 - x)e^{\eta/q}}\right) ; \quad g''(\eta) = \frac{1}{q} x(1 - x) \frac{e^{\eta/q}}{(x + (1 - x)e^{\eta/q})^2} ,$$

and
$$g'''(\xi) = \frac{1}{q^2}(-\tau + 3\tau^2 - 2\tau^3) , \text{ with } \tau = \frac{xe^{-\xi/q}}{1 - x + xe^{-\xi/q}} ,$$

where $\xi > 0$. In conclusion,
$$g(\eta) = g(0) + g'(0)\eta + g''(0)\frac{\eta^2}{2} + g'''(\xi)\frac{\eta^3}{6}$$
$$= \frac{1}{q} x(1 - x) \frac{\eta^2}{2} + g'''(\xi)\frac{\eta^3}{6} .$$

where $\xi \in [0, \eta]$. At this point we can examine the structure of the solution of (15) when $\eta \to 0$. Note that $q^*(x, \eta)$ necessarily satisfies
$$g(\eta) = \frac{1}{q^*(x, \eta)} x(1 - x) \frac{\eta^2}{2} + o(\eta^2) = \frac{\eta^2}{8} ,$$

implying that $q^*(x, \eta) \to 4x(1 - x)$ as $\eta \to 0$, proving the first statement in the lemma.

For the second statement in the lemma, one needs to more carefully control the error term $g'''(\xi)$. We begin by noting that $\tau \in [0, x]$. This implies that $-\tau + 3\tau^2 - 2\tau^3 \leq x(1 - x)$ (this can be checked by algebraic manipulation[1]). Therefore, $q^*(x, \eta) \leq q$, where $q$ is the solution of
$$\frac{1}{q} x(1 - x) \frac{\eta^2}{2} + \frac{1}{q^2} x(1 - x) \frac{\eta^3}{6} = \frac{\eta^2}{8} .$$

This is a simple quadratic equation in $q$, yielding the solution
$$q = 2x(1 - x) + \sqrt{(2x(1 - x))^2 + \frac{4}{3} x(1 - x)\eta} .$$

Although the above expression is a valid upper bound on $q^*(x, \eta)$, it is not a very convenient one. A more convenient upper bound can be obtained by noting that
$$2x(1 - x) + \sqrt{(2x(1 - x))^2 + \frac{4}{3} x(1 - x)\eta} \leq 4x(1 - x) + \eta/3$$
$$\iff (2x(1 - x))^2 + \frac{4}{3} x(1 - x)\eta \leq (2x(1 - x) + \eta/3)^2$$
$$\iff \frac{4}{3} x(1 - x)\eta \leq \frac{4}{3} x(1 - x) + \frac{\eta^2}{9} .$$

Thus, we have $q \leq 4x(1 - x) + \eta/3$, concluding the proof. □

---
[1] It suffices to check that the solutions in $\tau$ of $-1 + 3\tau - 2\tau^2 \leq 1 - x$, if they exist, satisfy $\tau \in [0, x]$.

# F   Proof of Theorem 3

In the proof of Theorem 3, we will require the following anytime version of Freedman's inequality:

**Lemma 3.** *Let $X_1, \ldots, X_n$ be a martingale difference sequence with respect to some filtration $\mathcal{F}_1 \subset \cdots \subset \mathcal{F}_n$ and with $|X_t| \leq b$ for all $t$ almost surely. Let $\Sigma_t^2 = \sum_{j=1}^t \mathbb{E}[X_j^2 | \mathcal{F}_{j-1}]$. Then, for any $\delta < 1/e$ and $n \geq 4$,*

$$
\mathbb{P}\left( \exists t \in [n] : \quad \sum_{j=1}^t X_j > 2 \max\left\{ 2\sqrt{\Sigma_t^2 \ln(1/\delta)}, b \ln(1/\delta) \right\} \right) \leq \ln(n)\delta .
$$

A proof of this result for the special case that $\mathcal{F}_t = \sigma(X_1, \ldots, X_t)$ can be found in [37, Lemma 3], which is an extended version of [38]. Their proof goes through unchanged for general filtrations.

We are now ready to prove Theorem 3. As aforementioned, we will prove the result under a more general condition than given in the theorem statement. Specifically, instead of assuming that the best expert in expectation is apparent from the first round, we only require the best expert to emerge after a time $\tau^*$. This condition is given in a precise form in (17). Clearly, the assumption stated in the main text implies that the condition in (17) holds with $\tau^* = 0$.

**Theorem 3** (with the best expert emerging after a time $\tau^*$). *Consider the label-efficient exponentially weighted forecaster from Theorem 2 with $q_t = \min(4A_{1,t}(1 - A_{1,t}) + \eta/3, 1)$ and any $\eta > 0$. Define $\tau$ as*

$$
\tau = \left\lceil \frac{48 \ln(1/\delta_2)}{\eta \Delta^2} + \frac{2}{\eta \Delta} \ln\left( \frac{N}{\eta} \right) \right\rceil . \tag{16}
$$

*Suppose that there exists a single best expert $i^*$ and a time $\tau^* \leq \tau$ such that, for all $i \neq i^*$,*

$$
\frac{1}{t} \sum_{j=1}^t \mathbb{E}_t[\ell_{i,j} - \ell_{i^*,j}] \geq \Delta > 0 \qquad \text{almost surely for } t \geq \tau^*. \tag{17}
$$

*Then, for any $n \geq 4$, the expected label complexity is at most*

$$
\mathbb{E}[S_n] \leq \frac{50}{\eta \Delta^2} \ln\left( \frac{N \ln n}{\eta} \right) + 3\eta n + 1 .
$$

*Proof.* Recall that the estimated losses are $\tilde{\ell}_{i,t} = \ell_{i,t} Z_t / q_t$. Let $\tilde{\Lambda}_t^i = \sum_{j=1}^t \tilde{l}_{i,j} - \tilde{l}_{i^*,j}$ denote the cumulative estimated loss relative to that of the best expert, and let $\tilde{\Lambda}_t^{\min} = \min_{i \neq i^*} \tilde{\Lambda}_t^i$.

Our argument separates the analysis in two regimes: Regime 1, where $t \leq \tau$, and Regime 2, where $t > \tau$. The specific choice of $\tau$ given above arises naturally later in the analysis. Regime 1, in which labels will be collected frequently, is expected to be relatively short, so we simply upper-bound $S_\tau \leq \tau$. It then remains to bound the number of collected labels in Regime 2. To this end, we need $\tau$ to be chosen such that

$$
q_t \leq \frac{4\eta}{3} \quad \text{for all } t > \tau \tag{18}
$$

with probability at least $1 - \delta_1$, where $\delta_1 \in (0, 1]$ will be fixed later. Let $\mathcal{E}$ denote the event that (18) holds. It then follows that the expected number of labels collected in Regime 2 is at most

$$
\mathbb{E}[S_n - S_\tau] = \mathbb{E}\left[ \sum_{t=\tau+1}^n q_t \mathbb{1}\{\mathcal{E}\} \right] + \mathbb{E}\left[ \sum_{t=\tau+1}^n q_t \mathbb{1}\{\bar{\mathcal{E}}\} \right] \leq \tfrac{4}{3}\eta n \Pr(\mathcal{E}) + n \Pr(\bar{\mathcal{E}}) \leq \tfrac{4}{3}\eta n + n\delta_1 .
$$

All together, we arrive at the following bound on the expected label complexity:

$$
\mathbb{E}[S_n] \leq \tau + \tfrac{4}{3}\eta n + n\delta_1 . \tag{19}
$$

It remains to verify that the choice of $\tau$ in (16) leads to $\delta_1$ being sufficiently small. To this end, let

$$
A_t^* := \frac{\sum_{i:f_{i,t}=f_{i^*,t}} w_{i,t-1}}{\sum_{i=1}^N w_{i,t-1}}
$$

denote the weighted proportion of experts that agrees with the best expert at time $t$. Then

$$A_{t+1}^* \geq \frac{w_{i^*,t}}{\sum_{i=1}^N w_{i,t}} = \frac{1}{1 + \sum_{i \neq i^*} e^{-\eta \tilde{\Lambda}_t^i}} \geq \frac{1}{1 + N e^{-\eta \tilde{\Lambda}_t^{\min}}}\,.$$

Note that $A_{1,t} \in \{A_{t+1}^*, 1 - A_{t+1}^*\}$. Consequently,

$$q_{t+1} \leq 4 A_{t+1}^*(1 - A_{t+1}^*) + \frac{\eta}{3} \leq 4(1 - A_{t+1}^*) + \frac{\eta}{3} \leq \frac{4}{1 + e^{\eta \tilde{\Lambda}_t^{\min}}/N} + \frac{\eta}{3}\,.$$

The desired condition from (18) is therefore satisfied if, for all $i \neq i^*$,

$$\tilde{\Lambda}_t^i \geq \frac{1}{\eta} \ln\left(\frac{N}{\eta}\right) \qquad \text{for all } t = \tau, \dots, n-1. \tag{20}$$

To study the evolution of $\tilde{\Lambda}_t^i$, we use a martingale argument. Consider any fixed $i \neq i^*$, and define the martingale difference sequence

$$X_t = -(\tilde{\ell}_{i,t} - \tilde{\ell}_{i^*,t}) + \mathbb{E}_t[\tilde{\ell}_{i,t} - \tilde{\ell}_{i^*,t}] = -\frac{(\ell_{i,t} - \ell_{i^*,t})Z_t}{q_t} + \mathbb{E}_t[\ell_{i,t} - \ell_{i^*,t}]\,.$$

Without loss of generality, we may assume that $\eta \leq 3$, because otherwise (19) holds trivially for any pair of $\tau$ and $\delta_1$. Then $q_t \geq \eta/3$, $|X_t| \leq 2/q_t \leq 6/\eta$, and

$$\mathbb{E}[X_t^2|\mathcal{F}_{t-1}] \leq \mathbb{E}\left[(\tilde{\ell}_{i,t} - \tilde{\ell}_{i^*,t})^2|\mathcal{F}_{t-1}\right] = \mathbb{E}\left[\frac{(\ell_{i,t} - \ell_{i^*,t})^2}{q_t}|\mathcal{F}_{t-1}\right] \leq \frac{3}{\eta}\,.$$

Hence, by Lemma 3, we have

$$\tilde{\Lambda}_t^i \geq \sum_{j=1}^t \mathbb{E}_t[\ell_{i,t} - \ell_{i^*,t}] - \max\left\{4\sqrt{\frac{3t}{\eta}\ln(1/\delta_2)}, \frac{12}{\eta}\ln(1/\delta_2)\right\} \qquad \text{for all } t \in [n]\,. \tag{21}$$

Using Assumption (17), it follows that

$$\tilde{\Lambda}_t^i \geq t\Delta - \max\left\{4\sqrt{\frac{3t}{\eta}\ln(1/\delta_2)}, \frac{12}{\eta}\ln(1/\delta_2)\right\} \qquad \text{for all } t \geq \tau^* \tag{22}$$

with probability at least $1 - (\ln(n)\delta_2)$ for any $\delta_2 \in (0, 1/e]$. By taking $\delta_2 = \min\{\delta_1/(N \ln n), 1/e\}$ and applying the union bound, we can make (22) hold for all $i \neq i^*$ simultaneously with probability at least $1 - \delta_1$. A sufficient condition for (22) to imply (20) is then to take

$$\tau = \left\lceil \frac{48 \ln(1/\delta_2)}{\eta \Delta^2} + \frac{2}{\eta \Delta} \ln\left(\frac{N}{\eta}\right) \right\rceil\,,$$

matching the definition in (16). We prove this claim in Lemma 4 below. Note that if our choice of $\tau > n$, then (19) still holds trivially, because $S_n \leq n$. Evaluating (19) with the given choice of $\tau$ and taking $\delta_1 = \eta$ (assuming $\eta < 1$, since (19) holds trivially if $\delta_1 \geq 1$), we arrive at the following bound:

$$\begin{aligned}
\mathbb{E}[S_n] &\leq \left\lceil \frac{48 \ln(1/\delta_2)}{\eta \Delta^2} + \frac{2}{\eta \Delta} \ln\left(\frac{N}{\eta}\right) \right\rceil + \tfrac{4}{3}\eta n + n\delta_1 \\
&\leq \frac{48 \ln(N \ln(n)/\delta_1)}{\eta \Delta^2} + \frac{2}{\eta \Delta} \ln\left(\frac{N}{\eta}\right) + \tfrac{4}{3}\eta n + n\delta_1 + 1 \\
&= \frac{48 \ln(N \ln(n)/\eta)}{\eta \Delta^2} + \frac{2}{\eta \Delta} \ln\left(\frac{N}{\eta}\right) + \tfrac{4}{3}\eta n + \eta n + 1 \\
&\leq \frac{50 \ln(N \ln(n)/\eta)}{\eta \Delta^2} + 3\eta n + 1\,,
\end{aligned}$$

thus completing the proof.

$\square$

**Lemma 4.** *Assume that*

$$\tilde{\Lambda}_t^i \geq t\Delta - \max\left\{4\sqrt{\frac{3t}{\eta}\ln(1/\delta_2)}, \frac{12}{\eta}\ln(1/\delta_2)\right\} \quad \textit{for all} \quad t \leq n .$$

*Then, with*

$$\tau = \left\lceil \frac{48\ln(1/\delta_2)}{\eta\Delta^2} + \frac{2}{\eta\Delta}\ln\left(\frac{N}{\eta}\right) \right\rceil ,$$

*we have*

$$\tilde{\Lambda}_t^i \geq \frac{1}{\eta}\ln\left(\frac{N}{\eta}\right) \quad \textit{for all} \quad t = \tau, \ldots, n-1 .$$

*Proof.* We will consider the two possible outcomes of the maximum separately. First, we consider the case where the first term is the maximum. Then, we need to find $\tau$ such that for $t \geq \tau$,

$$\tilde{\Lambda}_t^i \geq t\Delta - 4\sqrt{\frac{3t}{\eta}\ln(1/\delta_2)} - \frac{1}{\eta}\ln\left(\frac{N}{\eta}\right) \geq 0 .$$

A straightforward calculation shows that this is satisfied for

$$t \geq \frac{\left(\sqrt{48\ln(1/\delta_2) + 4\Delta\ln(N/\eta)} + 4\sqrt{3\Delta\ln(1/\delta_2)}\,\right)^2}{4\eta\Delta^2} .$$

Since $(a+b)^2 \leq 2a^2 + 2b^2$ for $a, b > 0$, this is satisfied given the simpler condition

$$t \geq \left\lceil \frac{48}{\eta\Delta^2}\ln(1/\delta_2) + \frac{2}{\eta\Delta}\ln\left(\frac{N}{\eta}\right) \right\rceil = \tau .$$

Next, we assume that the second term is the maximum. Then, we have

$$\begin{aligned}
\tilde{\Lambda}_t^i &\geq t\Delta - \frac{12}{\eta}\ln(1/\delta_2) \\
&\geq \tau\Delta - \frac{12}{\eta}\ln(1/\delta_2) \\
&= \left\lceil \frac{48\ln(1/\delta_2)}{\eta\Delta^2} + \frac{2}{\eta\Delta}\ln\left(\frac{N}{\eta}\right) \right\rceil \Delta - \frac{12}{\eta}\ln(1/\delta_2) \\
&\geq \frac{48\ln(1/\delta_2)}{\eta\Delta} + \frac{2}{\eta}\ln\left(\frac{N}{\eta}\right) - \frac{12\ln(1/\delta_2)}{\eta\Delta} \\
&\geq \frac{2}{\eta}\ln\left(\frac{N}{\eta}\right) ,
\end{aligned}$$

where we used the assumption that $t \geq \tau$. Thus, for this case, the desired condition holds with the specified $\tau$. Therefore, with the specified $\tau$, the desired statement holds for $t = \tau, \ldots, n-1$. □

## G   Experimental details

In this section, we describe the simulation study in Section 6 in detail. The full code, which can be executed in less than one hour on an M1 processor, is provided in the supplementary material.

We consider a sequential prediction problem with $n = 50000$ total rounds and $N = 225$ experts. The number of experts is chosen to be higher than $\sqrt{n}$ and odd. Then, we repeat the following simulation for 500 independent runs. First, we generate $n$ independent features $\{X_i\}_{i\in[n]}$ from the uniform distribution on $[0, 1]$. Then, for each feature $X_i$, we randomly generate a label $Y_i$, where the probability $\mathbb{P}(Y_i = 1 | X_i = x) = \zeta(x)$, where

$$\zeta(x) = \frac{1}{2} + \frac{1}{2}\text{sign}(x - 1/2)|x - 1/2|^{\kappa-1} .$$

Note that the optimal prediction of the label is simply $\mathbb{1}\{x \geq 1/2\}$. We run simulations both for $\kappa = 1.5$ and $\kappa = 2$.

We set the experts to be threshold classifiers, with thresholds uniformly spaced across $[0, 1]$. Specifically, for all $i \in [N]$ and $t \in [n]$,

$$f_{i,t} = \mathbb{1} \left\{ X_t \geq \frac{i-1}{N-1} \right\} .$$

In order to optimize the regret bound in (7), we set $\eta = \sqrt{8 \ln(N)/n}$. For each time $t \in [n]$, we consider two different weight vectors: $\{w_{i,t}^P\}_{i \in [N]}$, corresponding to the passive, full-information forecaster, and $\{w_{i,t}^A\}_{i \in [N]}$, corresponding to the active, label-efficient forecaster. The weight for each expert for both forecasters is uniformly initialized as $w_{i,0}^P = w_{i,0}^A = 1/N$.

Then, for each timestep $t \in [n]$, we proceed as follows. First, we compute the probability of prediction the label 1 for each forecaster, given by $p_t^P$ for the full-information forecaster and $p_t^A$ for the label-efficient forecaster, as

$$p_t^P = A_{1,t}^P = \frac{\sum_{i=1}^N w_{i,t-1}^P f_{i,t}}{\sum_{i=1}^N w_{i,t-1}^P} , \qquad p_t^A = A_{1,t}^A = \frac{\sum_{i=1}^N w_{i,t-1}^A f_{i,t}}{\sum_{i=1}^N w_{i,t-1}^A} ,$$

where the expert predictions are computed based on $X_t$. Here, the weighted proportion of experts that predict the label 1 are given by $A_{1,t}^P$ for the full-information forecaster and $A_{1,t}^A$ for the label-efficient forecaster. On the basis of this, each forecaster issues a prediction for the label, given by $\hat{y}_t^P \sim \mathrm{Ber}(p_t^P)$ for the full-information forecaster and $\hat{y}_t^A \sim \mathrm{Ber}(p_t^A)$ for the label-efficient one.

Finally, the weights for each forecaster are updated as follows. For the full-information forecaster, the weight assigned to each expert is updated as

$$w_{i,t}^P = w_{i,t-1}^P \, e^{-\eta \ell_{i,t}} , \tag{23}$$

where $\ell_{i,t} = \mathbb{1}\{f_{i,t} \neq Y_t\}$. In order to determine whether the label-efficient forecaster observes a label or not, we compute $q_t = \min(4A_{1,t}^A (1 - A_{1,t}^A) - \eta/3, 1)$ and generate $Z_t \sim \mathrm{Ber}(q_t)$. Then, the weights of the label-efficient forecaster are updated as

$$w_{i,t}^A = w_{i,t-1}^A \, e^{-\eta \frac{\ell_{i,t} Z_t}{q_t}} . \tag{24}$$

Hence, if $Z_t = 0$, the weights of the label-efficient forecaster are not updated, since the label is not observed.

After all $n$ rounds have been completed, we compute the regret of the full-information forecaster $R_n^P$ and the label-efficient forecaster $R_n^A$ as

$$R_n^P = \sum_{t=1}^n \mathbb{1}\{\hat{y}_t^P \neq Y_t\} - \mathbb{1}\{y_t^* \neq Y_t\} , \qquad R_n^A = \sum_{t=1}^n \mathbb{1}\{\hat{y}_t^A \neq Y_t\} - \mathbb{1}\{y_t^* \neq Y_t\} .$$

Here, for each $t \in [n]$, the optimal prediction is given by $y_t^* = \mathbb{1}\{X_t \geq 1/2\}$. Furthermore, we compute the label complexity $S_n$ as

$$S_n = \sum_{t=1}^n Z_t .$$

The results of these simulations are presented in Figure 2 and 3.

