# OpenReview forum: "Adaptive Selective Sampling for Online Prediction with Experts"
_NeurIPS.cc/2023/Conference — NeurIPS 2023 poster_

### Official Review · Reviewer_naDi · 2023-07-02

**Soundness:** 4 excellent
**Presentation:** 3 good
**Contribution:** 3 good
**Rating:** 7
**Confidence:** 4

**Summary:**

This paper presents an adaptive label-efficient forecasting technique for online binary prediction with expert advice. The proposed approach implements a label querying probability that is a function of the observed scenario, rather than based on pessimistic conditions. This enables the method to adapt, i.e., have lower label complexity, to benign environments while remaining robust to adversarial ones, unlike prior approaches in this label-efficient forecasting. Sharp analyses of the regret and label complexity and results on synthetic scenarios are provided in support of the method.

**Strengths:**

* The paper is very well-written and organized.
* The introduced algorithm is novel and intuitive. The regret and label complexity analyses of the approach seem sound.
* The method remedies the shortcoming (lack of adaptivity) of prior label-efficient prediction approaches. This enables it to query fewer labels in benign scenarios while remaining robust to adversarial ones.
* The authors present empirical evaluations that demonstrate the effectiveness of the method.


**Weaknesses:**

* The method only applies to binary prediction tasks with zero-one loss.
* Additional details on prior work on label-efficient prediction would be helpful in contextualizing the benefit (adaptivity) of the proposed approach.


**Questions:**


1. The authors mention that the same inductive approach can be used for general loss functions and $y_t \in [0,1]$, but that the expression for the selective sampling probability would be really complicated. Do the authors conjecture that an efficiently-computable upper bound for the complicated function can be found as was done for the binary zero-one loss case?
2. The authors mention active learning as a possible application in the introduction. Could this method (or its generalization) be applied to, e.g., a simple active learning scenario with a ResNet on CIFAR10?


**Limitations:**

Yes.

---

> ### Author Rebuttal · Authors · 2023-08-05
>
> We thank the reviewer for their careful reading and helpful comments.
>
> *Regarding general losses and predictions:*
>
> As you mention, applying the same approach to general losses would quickly lead to complications, but generalizing the analysis is an intriguing direction.
> Please see Point 2 in the global response for more details.
>
> *Regarding prior work:*
>
> Please see Point 3 in the global response. We will expand the discussion of related work in the paper along these lines.
>
> *Regarding active learning for e.g. CIFAR10:*
>
> This is a potential application of our approach.
> Given an ensemble of $N$ ResNets initialized with random weights, one could view them as experts and use the label-efficient forecaster to determine whether or not a given label should be observed, and used for training the ResNets.
> In order for the approach to work off-the-shelf, one would need to consider a binary classification version of CIFAR10.
> Extending the applicability of our results and using them in this way is an intriguing direction.

---

> > ### Comment · Reviewer_naDi · 2023-08-10
> >
> > Thank you for your response. I've read the authors' rebuttal and the other reviews and will maintain my position favoring acceptance (7).

---

> > > ### Author Response · Authors · 2023-08-19
> > >
> > > Thank you for your response!

---

### Official Review · Reviewer_G2NF · 2023-07-02

**Soundness:** 3 good
**Presentation:** 3 good
**Contribution:** 3 good
**Rating:** 8
**Confidence:** 4

**Summary:**

The paper considers a binary prediction game on $0-1$ loss, it proposed efficient sampling scheme via an modification of the exponentiated weight forecaster, which selectively acquire labels $y_t$ based on $Ber(q_t)$, where the design of $q_t$ is correlated to the disagreement among experts’ predictions at each round for the exponentiated weight forecaster.

Ultimately, the proposed algorithms achieves the regret of the exponentiated weight forecaster $O(\ln N \sqrt{ n} )$ over time horizon $n$ and $N$ number of experts, the design labelling acquisition parameter $q_t$ resultant to a labelling complexity of $O( \frac{\sqrt{n}}{\Delta^2} )$, where $0 < \Delta \le \mathbb{E} [\ell_{t,i} - \ell_{t,i^{\ast}}], \forall t \in [n], i \neq i^{\ast}$ represents the lower bound expected loss gap comparing to the optimal expert with index $i^{\ast}$, which also signifies the difficulty of identifying the optimal experts.

**Strengths:**

The paper is well written and easy to follow. It first introduces the intuition of sampling parameter $q_t$ if there is a perfect expert, then extended to general case. The paper also provided an graphical illustration on the lower bound of $q_t$ which matches with expectation.

The result is novel comparing to previous result in two folds. (1) There is no assumption on how $y_t$ is generated. The proposed algorithm is able to attain the $O(\ln N \sqrt{n})$ regret without less labels. (2) In contrast to previous work on sampling by disagreement, the label complexity can be quantified as $O(\frac{\sqrt{n}}{ \Delta^2})$.

$q_t$ is easy to compute, numerical experiment with respect to time horizon $n$ shows the expected regret and expected number of labels which matches with theoretical results. Experiment with respect to number of label (number of weights update) shows labelling efficient algorithm proposed in the paper matches the minimax rate in active learning asymptotically.




**Weaknesses:**

It seems that the assumption $0 > \Delta \ge \mathbb{E} [\ell_{t,i} - \ell_{t,i^{\ast}}], \forall t \in [n], i \neq i^{\ast}$ is required only for bounding the labelling complexity, ( in order to track how $q_{t+1}$ evolves in line 493), without this assumption the regret still holds.

I am a bit concern such assumption is very strong, it generally asks the best expert $i^{\ast}$ is wining over every other expert at every round. In addition, the assumption is $\ell_{t,i} \in \{ 0, 1\}$, if we are at a specific iteration $t$ where $\ell_{t,i^{\ast}} = 1$, then what values  can $\ell_{t,i}, \forall i \neq i^{\ast}$ take in order to satisfy the condition for a strictly positive $\Delta$?

Whether labelling complexity can be bounded without this assumption?


**Questions:**

Appendix C:
From line 437 to line 438, whether intermediate steps could be further elaborated. I am concerned about the proof, which seems primarily confused on the relationship of combining minimum values: $\min_i (a_i + b_i)$ VS $\min_i(a_i)+ \min(b_i)$.

My understanding is that from the last equation line of line 437, we have
$$ L_{n |1}^{w_{.,0} }  \le \frac{n \eta }{ 8 }  \underbrace{  - \frac{\eta}{8 } + A_{1,1} +  (1 - 2 A_{1,1})y_1  + \mathbb{E} \left[ \min_{i} \left( L_{i, n} + \frac{\ln w_{i,0}}{\eta} + \frac{q_1}{\eta} ( \ln w_{i,0} - \ln w_{i,1} )  - \ell_{i,1}  \right)   \right]}_{A} $$

To show the desired result claimed in the theorem, we aim to show
$$ A \le \mathbb{E} \left[ \min_{i} \left( L_{i,n} - \frac{\ln w_{i,0}}{\eta} \right) \right] $$

That is to show
$$  A_{1,1} +  (1 - 2 A_{1,1})y_1  + \mathbb{E} \left[ \min_{i} \left( L_{i, n} + \frac{\ln w_{i,0}}{\eta} + \frac{q_1}{\eta} ( \ln w_{i,0} - \ln w_{i,1} )  - \ell_{i,1}  \right)   \right]  \le  \frac{\eta}{8 } + \mathbb{E} \left[ \min_{i} \left( L_{i,n} - \frac{\ln w_{i,0}}{\eta} \right) \right] $$

From the claim in line 438, it seems we need to show above inequality should hold even without expectation, that is:
 $$  A_{1,1} +  (1 - 2 A_{1,1})y_1  + \min_{i} \left( L_{i, n} + \frac{\ln w_{i,0}}{\eta} + \frac{q_1}{\eta} ( \ln w_{i,0} - \ln w_{i,1} )  - \ell_{i,1}  \right) -  \min_{i} \left( L_{i,n} - \frac{\ln w_{i,0}}{\eta} \right)      \le  \frac{\eta}{8 }  $$

It seems the paper used
$$\min_{i} \left( L_{i, n} + \frac{\ln w_{i,0}}{\eta} + \frac{q_1}{\eta} ( \ln w_{i,0} - \ln w_{i,1} )  - \ell_{i,1}  \right) -  \min_{i} \left( L_{i,n} - \frac{\ln w_{i,0}}{\eta} \right) \le \min_{i} \left(  \frac{q_1}{\eta} ( \ln w_{i,0} - \ln w_{i,1} )  - \ell_{i,1}  \right) $$
which doesn't seem to be true. These intermediate steps are just my personal guess. I wonder whether details to understand those two lines in the manuscript can be further explained.




**Limitations:**

yes

---

> ### Author Rebuttal · Authors · 2023-08-05
>
> We thank the reviewer for their careful reading and helpful comments.
>
> *Regarding the assumption that $\Delta>0$:*
>
> Please see Point 1 in the global response, where we also highlight the more general assumption in Appendix F.
> Note that, under the assumption stated in the main paper, it is allowed
> that $\ell_{i^*,t}=1$ with positive probability, as long as $E_{t}[\ell_{i^*,t}]<1$.
> However, under the more general assumption in Appendix F, we even allow $E_t[\ell_{i^*,t}]=1$ for some rounds, as long as the best expert performs well in sufficiently many other rounds.
> Extending the label complexity analysis to even more general settings is
> interesting future work, but as mentioned in the paper, it is
> unavoidably linear in $n$ in the worst case (see also Point 3 in the global response).
>
> *Regarding line 437 to 438:*
>
> As you correctly observe, there is a (minor) issue with the proof as
> stated. Thanks a lot for catching this! It can be fixed as follows:
>
> In order to prove the desired result, it is enough to show that
>
> $$\bar L_{n |1}^{w_{.,0} } \leq \mathbb{E}\left[ \min_{i\in[N]} \left( \ell_{i,1} + L_{i,2:n} - \frac{\ln w_{i,0}}{\eta}\right) \right] + \frac{n \eta}{8}  = \mathbb{E}\left[ \left( \ell_{i',1} + L_{i',2:n} - \frac{\ln w_{i',0}}{\eta}\right) \right] + \frac{n \eta}{8} . $$
>
> Here, we let $i'$ denote the $\mathrm{argmin}$ of the right-hand side.
> While we had separated out $\ell_{i,1}$ in the submitted paper, this is not necessary.
> As shown after line 437, we have
>
> $$ \bar L_{n |1}^{w_{.,0} }  \leq A_{1,1}+(1-2A_{1,1})y_1 +\frac{(n-1)\eta}{8} + \mathbb{E}\left[\min_{i\in[N]} \left( L_{i,2:n}+q_1\frac{-\ln w_{i,1}}{\eta} + (1-q_1)\frac{-\ln w_{i,0}}{\eta}\right)\right] $$
>
> $$\leq A_{1,1}+(1-2A_{1,1})y_1 +\frac{(n-1)\eta}{8} + \mathbb{E}\left[ \left( L_{i',2:n}+q_1\frac{-\ln w_{i',1}}{\eta} + (1-q_1)\frac{-\ln w_{i',0}}{\eta}\right)\right] \ .$$
>
> In the last step, we used the fact that since the upper bound holds for the minimum $i$, it holds for $i'$ in particular.
> Now that everything has been recast in terms of $i'$, the rest of the argument follows as stated.
> To ensure that the bound in the theorem holds, it is thus sufficient to select $A_{1,1}$ such that it satisfies
>
> $$A_{1,1}+(1-2A_{1,1})y_1+\left(\frac{q_1}{\eta}\left(\ln w_{i',0}-\ln w_{i',1}\right)-\ell_{i',1}\right)\leq \frac{\eta}{8}\ ,$$
>
> after which the proof continues as before.
> Thank you for pointing this out!

---

> > ### Comment · Reviewer_G2NF · 2023-08-13
> >
> > Dear Author,
> >
> > Thank you very much for your response. I have read through global response which addressed the condition on $\delta$, and the interpretation (the expected cumulative loss gap growth). Also, thank you for addressing the proof from line 437 to 438.
> >
> > After reading all responses, I elevated my score from 7 to 8, since this is the first paper being able to quantify a $O(\sqrt{T})$ label complexity under the considered assumption given the best of my knowledge.

---

> > > ### Author Response · Authors · 2023-08-19
> > >
> > > Thank you for your response and updated score!

---

### Official Review · Reviewer_VgJf · 2023-07-07

**Soundness:** 4 excellent
**Presentation:** 4 excellent
**Contribution:** 4 excellent
**Rating:** 8
**Confidence:** 4

**Summary:**

This paper proposes an interesting novel approach to prediction with expert advise. In the standard prediction with expert advise setup, the learner receives experts' predictions, commits to its own and then sees the true outcome as produces by the (possibly adversarial) nature. Suppose that obtaining the true outcome is costly; do we really need to do this all the time? Clearly, if all (or most) experts agreed on the same prediction, the value of the true outcome for adjusting our trust in them is negligible; in the standard weight-based algorithms with multiplicative update the contribution of this round will simply be eliminated by normalisation.

The paper takes an algorithm from Cesa-Bianchi and Lugosi (exponential weighting with fixed $\eta$) and shows that its regret bound stands as it is if the true outcome is requested with certain probability.

It is then shown that the expectation of the number of outcomes actually requested is upper bounded by $3\eta T + O(\log\log T)$, so for small $\eta$ there is linear improvement in the number of requested outcomes. If $\eta$ is chosen to minimise the regret bound using prior knowledge of $T$, we take $\eta\propto 1/\sqrt{T}$ and thus the bound reduces to $O(\sqrt{T})$. This results holds under some conditions on experts' behaviour  though and they seem restrictive.

**Strengths:**

I think this is an interesting take on the well-known problem and should be published.

**Weaknesses:**

No obvious weaknesses.

**Questions:**

The authors chose to present the results in the probabilistic setup. Typically predicting $0$ or $1$ in a probabilistic setup is equivalent to predicting $\gamma\in [0,1]$ under a deterministic setup; the later seems more important to me (this is my judgement of course). Will the results of the paper stand?

I think the final version would benefit from a better discussion of the $O(T)$ lower bound, which is mentioned but not discussed in detail.

**Limitations:**

Yes

---

> ### Author Rebuttal · Authors · 2023-08-05
>
> We thank the reviewer for their careful reading and helpful comments.
>
> *Regarding the restrictiveness of the conditions for the label complexity bound:*
>
> While the assumed setting in Theorem 3 is relatively benign, it includes many relevant settings, and the results hold under the more lenient assumption of cumulative separation after a certain time.
> Please see Point 1 in the global response for more details.
>
> *Regarding extensions to predictions in $[0,1]$:*
>
> Extending our analysis to predictions in $[0,1]$ would be interesting, and we believe it may be possible to extend our approach.
> Please see Point 2 in the global response for more details.
>
> *Regarding the label complexity lower bound:*
>
> We will update the paper with further discussion of the lower bound.
> Please see Point 3 in the global response for more details.

---

> ### Comment · Reviewer_VgJf · 2023-08-11
>
> Many thanks to the authors for the response.
>
> I do think it should be straightforward to extend the result to \[0,1\], but regardless of this I am happy to keep my high evaluation of the paper.

---

> > ### Author Response · Authors · 2023-08-19
> >
> > Thank you for your response! After thinking a bit more about this, we agree that it is straightforward to extend our results to expert predictions in [0,1], while still keeping the label sequence and forecaster predictions binary (in {0,1}). Extending to more general forecaster predictions and losses does not seem quite as straightforward, although it may be doable with similar techniques.

---

### Official Review · Reviewer_1FCH · 2023-07-08

**Soundness:** 3 good
**Presentation:** 2 fair
**Contribution:** 3 good
**Rating:** 6
**Confidence:** 3

**Summary:**

This paper investigates the PEA problem in the context of online binary classification where the cost of obtaining labels for streaming data is high, necessitating selective label collection adaptively. To this end, the authors introduce a carefully designed label collection strategy based on the classical Hedge algorithm. The resultant label-efficient forecaster has a best-of-both-worlds theoretical guarantee. The authors further demonstrate that regret of their label-efficient forecaster asymptotically reaches the minimax rates of pool-based active learning.

**Strengths:**

* The paper is well-structured and easy to comprehend.
* The theoretical analysis provided in the paper appears to be solid and sound.

**Weaknesses:**

* The primary conclusion of the paper, Theorem 3, relies on a core assumption that there exists a unique optimal expert whose expected loss in each round surpasses all other experts by a specific margin $\Delta$. Is this assumption too strong? Are there real-world application scenarios that satisfy such an assumption? (To my knowledge, this assumption has only been used in the COLT'14 paper: A Second-order Bound with Excess Losses).

* Does this specific PEA setting investigated in the paper relates to the bandit setting? Both are concerned with identifying the optimal expert (arm). How then do the setting and techniques used in this paper differ or relate to those in bandit scenarios? Does the problem studied in this paper present novel challenges in comparison to the bandit setting?


* The paper states that the online prediction setting is similar to streaming active learning. If this is the case, should the numerical experiment compare the regret convergence rate of the proposed algorithm with that of the streaming active learning algorithm?


* Some writing aspects could be improved:

  * In line 227, the symbol $\tilde l$ is undefined.
  * In line 264, the regret bound (7) is termed "pessimistic" without explanation.

**Questions:**

See Weaknesses above.

**Limitations:**

The paper should further clarify the method and theory part, see Weaknesses above.

---

> ### Author Rebuttal · Authors · 2023-08-05
>
> We thank the reviewer for their careful reading and helpful comments.
>
> *Regarding the assumption in Theorem 3:*
>
> The assumed setting is relatively benign, but it does include many practical i.i.d. settings, and the stated results do hold under a more general assumption (as detailed in Appendix F).
> Please see the global response for more details, especially the point
> that our algorithm still works when the assumption is violated.
>
> *Regarding the relation to the bandit setting:*
>
> This is an interesting question.
> In the setting considered in the paper, we observe the losses of *all* experts when observing a label.
> In contrast, in the bandit setting, only the loss of the selected arm would be observed for each round.
> This would necessitate the forecaster to incorporate more exploration in its strategy, and the analysis of a label-efficient version seems like it would be very different from what is used in this paper, although some of the ideas may transfer.
>
> *Regarding the comparison to active learning:*
>
> Essentially, streaming active learning can be seen as a special case of label-efficient online prediction, where the environment draws i.i.d. features and labels, and the expert predictions are determined by fixed functions of the features.
> Note that this is precisely the setting that we consider in the numerical experiments.
> Thus, the comparison to batch active learning is a comparison between the following:
>
> 1. minimax optimal batch active learning,
>
> 2. streaming active learning based on our label-efficient algorithm for online learning.
>
> The fact that these match asymptotically is an indication that, for the specific setting under consideration, the attainable performance with batch active learning and streaming active learning coincide asymptotically, and specifically, this is achieved by using our proposed label-efficient predictor for streaming active learning.
> However, as stated in the paper, this is an empirical observation which we did not establish theoretically.
>
> *Regarding aspects of the writing:*
>
> We will rewrite Line 227 in terms of $\ell_{i,j}/q_{j}$ (to avoid needing
> to introduce the short-hand $\tilde \ell$) and we will clarify that (7)
> is ‘‘pessimistic'' because it is a worst-case bound for adversarial environments.

---

> > ### Comment · Reviewer_1FCH · 2023-08-16
> >
> > Thanks for your response, which has addressed my concerns and I will raise my score to 6. I don't have any further questions.

---

> > > ### Author Response · Authors · 2023-08-19
> > >
> > > Thank you for your response and updated score!

---

### Author Rebuttal · Authors · 2023-08-05

## Global response to all reviewers

We thank all reviewers for their careful reading and helpful comments.
We are happy that you consider the paper to be sound, novel, interesting, and well-written.

Below, we address three points that were raised by multiple reviewers:

*1. Regarding the assumption of a best expert with $\Delta>0$:*

The setting in Theorem 3, where there exists a single expert that is
best in expectation for all rounds, is satisfied for any static
stochastic environment with a unique minimizer, including many relevant
applications with i.i.d. data. The same assumption is standard in the
stochastic bandit setting, where it is nearly always assumed that there
exists a single best arm, and has also been used in the prediction with
expert advice setting. We therefore view it as relatively
uncontroversial. Moreover, note that our algorithm satisfies a
best-of-both-worlds result: it preserves regret guarantees, even if the $\Delta > 0$
assumption does not hold, but if the assumption does hold, the label complexity is much smaller than $n$. We further note that, as mentioned before the theorem statement, the same result holds under a more general assumption: the best expert does not need to surpass the others in each round.
Instead, it is sufficient that the normalized *cumulative* expected loss becomes separated by $\Delta$ sufficiently fast.
This is stated more precisely in Eq. (16) and (17) in Appendix F in the supplementary material.
Under the more general assumption, the ‘‘best'' expert is even allowed to be the worst for some rounds, as long as it performs well in sufficiently many other rounds.
Extending the label complexity analysis to even more general settings is interesting future work, but as mentioned in the paper, it is linear in $n$ in the worst case (Theorem 13 in [10], N. Cesa-Bianchi, G. Lugosi, and G. Stoltz: Minimizing regret with label efficient prediction).

*2. Regarding the extensions to general losses and predictions:*

While extending the results to losses and predictions in $[0,1]$ seems solvable by generalizing our approach, this will complicate the optimization problem in Eq. (13,14), leading to a more complicated (implicit) specification for $q_{t}$.
It may be the case that an altered technique based on the same ideas would allow for efficiently handling more general losses and predictions.
As we have not fully established the details, this should be taken with a grain of salt, but it is an interesting avenue for future work.

*3. Regarding the worst-case label complexity:*

The label complexity being linear in $n$ in the worst case follows from Theorem 13 in [10] (N. Cesa-Bianchi, G. Lugosi, and G. Stoltz: Minimizing regret with label efficient prediction).
To clarify this point, we will alter the beginning of Section 5 as follows in the revised paper:

___

*We now examine the label complexity, defined as $S_{n}\equiv \sum_{t=1}^{n} Z_{t}$.
In [Thm. 13, 10], it is shown that there exists a setting for which the expected regret of a forecaster that collects $m$ labels is lower-bounded by $cn\sqrt{\ln(N-1)/m}$ for some constant $c$. Hence, in the worst case, the number of collected labels needs to be linear in $n$ in order to achieve an expected regret that scales at most as $\sqrt n$. However, since $q^{\*}_t$ can be less than $1$, it is clear that the label-efficient exponentially weighted forecaster from Theorem 2 can collect fewer than $n$ labels in more benign
settings. To this end, we consider a scenario with a unique best expert,
which at each round is separated from the rest in terms of its expected
loss.*

---

### Decision · Program_Chairs · 2023-09-21

**Decision:**

Accept (poster)

**Comment:**

This work proposes label-efficient algorithms for binary prediction with expert advice. The algorithms employs a selective sampling scheme, and queries fewer labels in benign settings while having optimal worst-case regret guarantees. In the benign setting where an expert is strictly better than the rest, explicit label complexity is obtained. The paper is well-written and the proposed method is novel; while there were concerns about the assumption under which the explicit label complexity is obtained, there is a worst-case lower bound, and this work is the first to obtain a sublinear label complexity. I therefore recommend acceptance.